# Robust Hallucination Detection in LLMs via Adaptive Token Selection

**Mengjia Niu**[1,2]*, **Hamed Haddadi**[1], **Guansong Pang**[2]†
[1]Imperial College London, UK
[2]Singapore Management University, Singapore
m.niu21@imperial.ac.uk, h.haddadi@imperial.ac.uk, gspang@smu.edu.sg

## Abstract

Hallucinations in large language models (LLMs) pose significant safety concerns that impede their broader deployment. Recent research in hallucination detection has demonstrated that LLMs' internal representations contain truthfulness hints, which can be harnessed for detector training. However, the performance of these detectors is heavily dependent on the internal representations of predetermined tokens, fluctuating considerably when working on free-form generations with varying lengths and sparse distributions of hallucinated entities. To address this, we propose HaMI, a novel approach that enables robust detection of hallucinations through adaptive selection and learning of critical tokens that are most indicative of hallucinations. We achieve this robustness by an innovative formulation of the **Ha**llucination detection task as **M**ultiple **I**nstance (**HaMI**) learning over token-level representations within a sequence, thereby facilitating a joint optimisation of token selection and hallucination detection on generation sequences of diverse forms. Comprehensive experimental results on four hallucination benchmarks show that HaMI significantly outperforms existing state-of-the-art approaches. Code is available at https://github.com/mala-lab/HaMI.

## 1 Introduction

Recent progress in Large Language Models (LLMs) has demonstrated impressive capabilities across a wide range of applications. However, the ever-growing popularity of LLMs also gives rise to concerns about the reliability of their outputs [12, 22]. Some research has indicated that LLMs are susceptible to hallucinations, which can be described as unfaithful or incorrect generations [2, 18, 33]. This tendency not only impedes the broader applications of LLMs but also poses potential safety risks, especially in high-stake fields such as legal and medical services. Therefore, the reliable detection of hallucinations is critical for the safe deployment of LLMs.

Various approaches have been developed to detect hallucinations. Recent studies indicate that predictive uncertainty can serve as useful detection features [9, 15, 25], as predictions with low confidence often correlate with the presence of hallucinated content. Some research focuses on the evaluation of a single generation [6, 34] while some harness the information contained in multiple generations. Compared to single-generation methods, approaches utilising multiple generations demonstrate greater effectiveness since they capture a broader set of hallucination signals. Such signals include semantic consistency [17] for the repeated queries, as well as behaviour inconsistency when LLMs are presented with a set of unrelated follow-up questions [38] after the initial question. Nevertheless, these methods are mainly based on the final generation output, rendering them ineffective in leveraging important semantics in the internal representations.

---

*The work was done when Mengjia Niu visited Singapore Management University.
†Corresponding author: G. Pang (gspang@smu.edu.sg)

39th Conference on Neural Information Processing Systems (NeurIPS 2025).

In parallel, another line of research focuses on the utilisation of the internal representations of LLMs for hallucination detection [5, 32]. These internal representations can encode information about the truthfulness direction of the generations. Motivated by this, great efforts have been made to explore the characteristics of these representations, most commonly by training a binary classifier on them. Various supervision signals, including accuracy labels [30], converted semantic entropy labels [26], and eigenvalue-related labels [14], have been harnessed to train the classifier for detection tasks. One major challenge for these methods is that a majority of tokens in an incorrect/hallucinated response may not contribute to truthfulness. To address this issue, most methods utilise predetermined tokens, such as the first generated token, the last generated token, or the one before the last [7, 21, 52]. However, the exact location of the most indicative tokens for hallucination can vary significantly, as illustrated in Figure 1, since the generation responses are of free-form with varying lengths and have sparse distributions of hallucinated entities. As a result, they can overlook important tokens where hallucinated information is actually concentrated.

To address this challenge, we propose **HaMI**, namely, **Ha**llucination detection as **M**ultiple **I**nstance learning, a novel approach that jointly optimises token selection and hallucination detection in an end-to-end fashion. The joint optimisation enables adaptive token selection from internal state representations for stable and accurate hallucination detection on generation responses of varying length. In HaMI, we reformulate the task as a multiple instance learning (**MIL**) problem [10], where each response sequence is treated as a bag of token instances, with a bag-level label as either hallucinated (positive) or trustworthy (negative), and the objective becomes binary classification of the token bags. This way takes advantage of the fact that only a few token instances in the positive bag are positive since hallucinated content typically manifests in only a small subset of tokens within a sequence, whereas all token instances in the negative bag are always negative. In doing so, the MIL approach enables the exploitation of the hallucination labels at the sequence (bag) level to adaptively select the most responsive tokens for sequence-level hallucination detection.

Figure 1: Tokens that contain the most sufficient information related to correctness may appear at various positions within the sequence.

To be more specific, LLMs are first prompted to generate response sequences with varying length. The MIL-driven hallucination detector is then optimised in HaMI to assign hallucination scores to all individual token instances and adaptively select the most indicative tokens in both positive and negative bags for the sequence-level prediction. The optimisation results in a detector that can distinguish the most positive hallucinated token instances from the hard negative token instances (*i.e.,* the tokens that have the highest hallucination scores in a negative bag). Additionally, recognising that predictive uncertainty serves as an important indicator of correctness, we further propose a representation enhancement module in HaMI, where we integrate multiple levels of uncertainty information into the original representation space for more effective training of our HaMI detector.

In summary, our contributions are as follows:

- We propose HaMI, a novel MIL-based framework for hallucination detection, which enables an end-to-end joint optimisation of token selection and hallucination detection. This warrants adaptive selection of hallucinated tokens that are optimal for the learned detector, effectively mitigating the performance instability on response generations of varying length. To our best knowledge, this is the first approach allowing such a joint optimisation.

- We further introduce a module that incorporates internal representations with uncertainty scores to provide more indications of hallucination for the joint optimisation in HaMI.

- Comprehensive empirical results with widely adopted LLMs on four popular benchmark datasets show that HaMI can significantly outperform state-of-the-art (SotA) methods.

## 2 Related Work

The term hallucination, under the closed-book setting, can refer to unfaithful or fabricated generations [22, 50]. Having gained wide interest, hallucination detection is crucial for LLMs to maintain high

reliability in various specific tasks. These methods can be categorised into two main lines: uncertainty measurement and internal representation analysis.

**Uncertainty Measurement-based Methods.** Uncertainty measurement has been widely explored for hallucination detection. Some research focuses on token-level uncertainty [15, 40, 46] with the assumption that low predictive logits or high entropy over tokens' predictive distribution indicate a high possibility of hallucination. Some studies target sentence-level uncertainty to capture a more holistic view of the whole response. The uncertainty can be measured by (weighted) aggregation of all token logits, such as Perplexity [42], Mars [6] and G-NLL [3], or obtained by directly instructing LLMs to express the truthfulness of their generations with simple but effective prompts [25, 31, 34, 51].

To further enrich semantic understanding and improve the detection accuracy, researchers also explore LLMs' behaviour among multiple generations [13, 36]. For example, Farquhar *et al.* [17] propose Semantic Entropy as an evaluation of semantic consistency among multiple generations for hallucination detection, while Pacchiardi *et al.* [38] present that behaviour inconsistency over predefined follow-up unrelated questions can also indicate the correctness of the response to the initial question. Although effective, the performance of these methods relies on external tools and ignores the important semantics embedded in the internal representations of LLMs.

**Internal Representation-based Methods.** Recently, a branch of work suggests that the internal representations of LLMs encode more knowledge than they express and can reveal truthfulness direction [8, 19, 44, 49]. The majority of this line employs probes [4] to better understand head-wise or layer-wise representations and predict the correctness of generations [9, 11, 30, 35]. Recent research extends these methods by proposing new supervision signals, such as an automated membership estimation score presented in HaloScope [14] and the aforementioned semantic entropy value [26], which has been proven to be preferable to accuracy labels for supervised training. Most of these works leverage predefined token representations, but the truthfulness information is concentrated in specific tokens. Some research attempts to prompt an LLM to find the "exact token" in the sequence [37], but the reliability of detection is greatly influenced by the capability of the employed LLM. Unlike existing methods that separate token selection and hallucination detection into two stages and require external assistance, we propose to train a ranking model to automatically select the critical tokens for hallucination detection in an end-to-end manner.

## 3 Preliminaries

Given a sequence of input tokens $\mathbf{x} = \{x_1, x_2, \ldots, x_m\}$ consisting of a specific question $q$, an LLM will generate a sequence of tokens $\mathbf{y} = \{y_1, y_2, \ldots, y_t\}$. Generally, each token $y_{i \in \{1,2,\ldots,t\}}$ is decoded from the next predictive distribution over the model's vocabulary set $\mathcal{V}$, formulated as $y_i = \arg\max_{y \in \mathcal{V}} P(y|y_{<i}, \mathbf{x})$, and the predictive probability is denoted by $p_i$ for short. By accessing the internal states of the model, we can extract the internal representation $h_{i,l} \in \mathbb{R}^d$ at layer $l$ for each token $y_i$, where $d$ is decided by the dimensions of the internal states of the LLM. The correctness of generated response is evaluated by GPT-4.1 [1] with label $z \in \{0, 1\}$.

Given a datset $D = \{(q_n, a_n)\}_{n=1}^N$, where $\{q_n\}_{n=1}^N$ and $\{a_n\}_{n=1}^N$ are questions and ground-truth answers respectively, LLMs will generate answers decoded from a token list $\mathbf{y}_n$ accompanied with predictive probabilities, hidden representations $\mathbf{h}_n$ and correctness labels $z_n \in \{0, 1\}$. The representation space for all samples is denoted as $\mathcal{H} = \{\mathbf{h}_n\}_{n=1}^N$ Our MIL-based hallucination detection is to identify the most representative tokens in incorrect responses and the hard negative tokens (*i.e.,* the most likely hallucinated tokens within correct responses) that are optimal for training the subsequent detector, *i.e.,* a joint optimisation of token selection and hallucination detection.

## 4 Methodology

Our proposed HaMI aims to distinguish hallucination-free and hallucination-containing text generations by adaptive token selection that jointly optimises token selection and hallucination detection. Additionally, we introduce a predictive uncertainty-based module to integrate more hallucination features in HaMI and enhance its discriminative capability. The overall framework is presented in Figure 2. Below we introduce these two modules in detail.

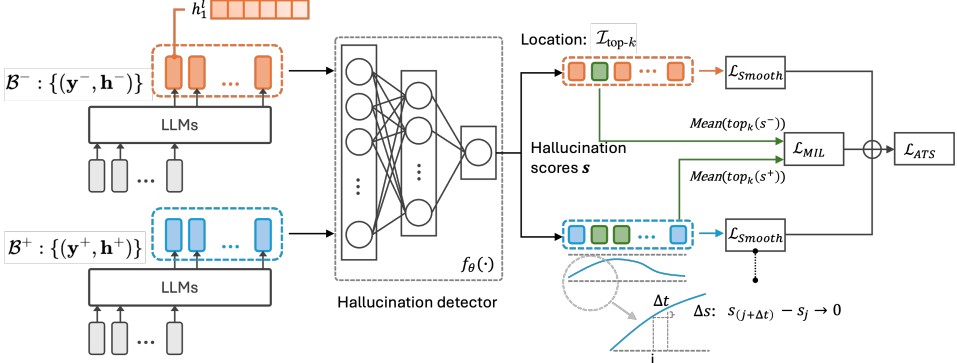

Figure 2: The framework of our proposed HaMI. The LLM is prompted to generate answer tokens accompanied by token representations $h_i$. The network receives sequences of token representations for both positive $\mathcal{B}^+$ and negative $\mathcal{B}^-$ bags as inputs at the training stage. The hallucination detector assigns a hallucination score to each token instance. We choose the top $k$ largest scores from both bags, subsequently maximising the discriminative margin between them by minimising a MIL loss as described in Eq. 2. Given the sequential nature of the generations, a constraint on the smoothness of hallucination scores of adjacent tokens is also added to HaMI.

## 4.1 MIL-driven Adaptive Token Selection

For a given sentence, its correctness is often determined by just a few words, such as noun entities [12], suggesting that truthfulness information may be encoded in the internal representations of specific tokens. Nevertheless, the correctness label is typically assigned to the entire sequence. The key idea of our approach is to adaptively identify the most salient tokens, upon which we can train a reliable hallucination detector. To this end, we introduce Multiple Instance Learning (MIL) to this domain.

In MIL, instead of finding the classification boundary for samples with different identities, it tries to distinguish between the hard instance from sample bags of various categories. To make it clear, there are positive bags and negative bags containing multiple instances. All the instances in negative bags are negative while only a few instances in positive bags are positive. MIL aims to separate the positive instances and the hard negative instances from negative bags. This objective aligns with our assumption that there are only several tokens containing information of hallucination.

Therefore, we reformulate hallucination detection with adaptive token selection as an MIL problem. Particularly, the generated sequence can be regarded as a bag of tokens. Generations without hallucinations are labelled as negative bags $\mathcal{B}^-$ (label '0'), while those with hallucinations are labelled as positive bags $\mathcal{B}^+$ (label '1'). The representations of the positive and negative sequences are denoted as $\mathbf{h}^+$ and $\mathbf{h}^-$ respectively and $\mathbf{h} = \{h_i\}_{i=1}^t$. Intuitively, the detector is expected to assign higher scores to token instances from the positive bag compared to those from the negative bag. Given the hallucination sparsity insight mentioned before, we choose the token instances with the top $k$ highest hallucination scores as the salient tokens in each sequence, which can be defined as

$$\mathcal{I}_{\text{top-}k} = \{i_1, i_2, \ldots, i_k\} \quad \text{s.t.} \quad \begin{cases} f_\theta(h_{i_1}) \geq f_\theta(h_{i_2}) \geq \cdots \geq f_\theta(h_{i_k}) \\ f_\theta(h_{i_k}) \geq f_\theta(h_{i_j}), \quad \forall i_j \notin \mathcal{I}_{\text{top-}k} \end{cases} \quad (1)$$

where $i_k \in \{1, 2, \cdots, t\}$, and $f_\theta$ is the hallucination detector we aim to learn with the parameters $\theta$, as presented in Figure 2. As such, we are able to locate the most hallucinated token instances in the positive bag and the top hard negative token instances resembling a hallucination in the negative bag as the salient tokens. Our approach then seeks to distinguish between positive and negative bags by maximising the distance between the selected token instances from these two categories in the representation space. The training MIL objective can be formulated as follows:

$$\mathcal{L}_{MIL} = 1 - \|\frac{1}{k} \sum_{i^+ \in \mathcal{I}_{\text{top-}k}^+} f_\theta(h_{i^+})\|_2 + \|\frac{1}{k} \sum_{i^- \in \mathcal{I}_{\text{top-}k}^-} f_\theta(h_{i^-})\|_2, \quad (2)$$

where $h_{i+}$ is selected from $\mathbf{h}_n^+ \in \mathcal{B}^+$ and $h_{i-}$ is from $\mathbf{h}_n^- \in \mathcal{B}^-$, and $\mathcal{I}_{\text{top-}k}^+$ and $\mathcal{I}_{\text{top-}k}^-$ are index sets for tokens in sequences from $\mathcal{B}^+$ and $\mathcal{B}^-$ respectively. In doing so, the adaptive token selection can mitigate the limitations caused by predefined token location.

Note that the next token is conditioned on all previous tokens, so the token generation process contains a sequential nature. As depicted in Figure 2, on the same side of the peak, the hallucination scores for two adjacent tokens tend to be more similar. Therefore, we exploit sequential smoothness via the following loss:

$$\mathcal{L}_{Smooth} = (f_\theta(h_i) - f_\theta(h_{i-1}))^2, \tag{3}$$

where $i \in \{2, \cdots, t\}$. Through this, we aim to ensure the consistency of the hallucination scores of neighbouring tokens. The effectiveness of $\mathcal{L}_{Smooth}$ is discussed in Appendix B in detail. Finally, HaMI is optimised by minimising the following overall loss:

$$\mathcal{L}_{ATS} = \mathcal{L}_{MIL} + \mathcal{L}_{Smooth}. \tag{4}$$

## 4.2 Augmenting Internal State Representations with Predictive Uncertainty

Many studies have demonstrated that both uncertainty measurements and internal representations can capture truthfulness-related information, motivating us to explore the synergy of them to enhance hallucination detection. Moreover, internal states encode broader linguistic patterns than truthfulness, so it is intuitive to investigate whether integrating uncertainty can act as a source of truthfulness to amplify the internal representations. In this work, we seek to determine whether incorporating uncertainty metrics into the original token representation space can enhance their discriminative capability in identifying hallucinations.

The collection of predictive uncertainty measurements can be categorised into three levels: **i)** *token-level uncertainty*, *i.e.,* predictive probabilities:

$$P_{\text{uncertainty}}^t = P(y|y_{<i}, \mathbf{x}), \tag{5}$$

where $\mathbf{x}$ are prompt tokens, $y$ are generated tokens and $i \in \{1, 2, \cdots, t\}$; **ii)** *sentence-level perplexity*, which is monotonically related to the mean of the negative log-likelihood of the sequence:

$$P_{\text{uncertainty}}^s = -\frac{1}{T} \sum_{t=1}^{T} \log P(y|y_{<i}, \mathbf{x}), \tag{6}$$

and **iii)** *semantic consistency across multiple samples*, where the uncertainty value can be quantified by the number of semantic-equivalence generations over the whole generations based on the entailment results. We choose Semantic Entropy here [17]. Specifically,

$$P_{\text{uncertainty}}^c = \sum_{1}^{M} \frac{I_c = c_m}{M}, \tag{7}$$

where $M$ is the total number of generations, $c_m$ is the identified cluster and $I_c$ is the assigned cluster identity of a given sample.

These uncertainty metrics can be directly leveraged for hallucination detection, and on average, the semantic consistency outperforms the other two metrics. However, it requires more computational cost while the other two metrics can obtained from just one generation as long as we can access the internal states of LLMs. To augment the internal representations with the uncertainty information, we define the final input representation for each token as follows:

$$\mathbf{h}' = (1 + \lambda \cdot P_{\text{uncertainty}}) \cdot \mathbf{h}, \tag{8}$$

where $P_{\text{uncertainty}}$ can be instantiated by one of the above three measurements, and $\lambda$ is used to control the impact of uncertainty metrics. The improvements gained by various uncertainty measurements and the choice of $\lambda$ are evaluated and discussed in Section 5.3.

## 5 Experiments

**Datasets and Models.** We evaluate our method on four popular benchmark datasets across a range of question-answering (QA) domains, including (1) Trivia QA [24], a relatively complicated

confabulation QA datasets; (2) Stanford Question Answering Dataset (SQuAD for short) [41], which is based on Wikipedia and generated by humans through crowdsourcing; (3) Natural Questions (denoted as NQ) [28], containing search information from real users on Google search and (4) a biomedical QA corpus BioASQ [27]. For each dataset, we randomly extract $2,000$ QA pairs for training and $800$ pairs for testing. For multi-sampling approaches, we prompt LLMs six times for each question for the test set generation. Following [17], we have a total of $500$ questions randomly sampled from the generated test set, with approximately $400$ questions retained as the final inputs at the testing stage for evaluation. The refined set from the remaining $300$ QA pairs is used as a validation set for selecting the optimal layer for representation extraction. We employ representative open-sourced LLMs, LLaMA [16, 47] and Mistral family [23], and evaluate our approach on LLaMA-3.1-8B and Mistral-Nemo-Instruct (12B), as well as a larger version of LLaMA, namely LLaMA-3.3-Instruct-70B, which is deployed with 8-bit quantisation [48]. All answers are generated under the temperature of $0.5$ and with context-free zero-shot prompts. Please see more details for the setup in Appendix A.2.

**Baselines.** We compare our method with a series of state-of-the-art (SotA) methods, covering both uncertainty-based methods and internal representation-based methods. The uncertainty-based methods include: **p(True)** [25], that asks LLMs to express the correctness of given answers themselves; **Perplexity** [42], which is based on Eq. 6; **Semantic Entropy (SE)** [17], semantic-equivalence measurement across multiple samples, where GPT-3.5 is used for entailment evaluation; **MARS** [6], that performs a weighted aggregation of token logits, and its enhanced version (**MARS-SE**), augmented with semantic entropy. For internal representation-based methods, probing classifiers with various supervision signals are included: **CCS** [9], utilising contrast-consistent discovering for correctness probability measurements; **SAPLMA** [5], trained with correctness labels; **HaloScope** [14], which proposes an automated membership estimation score and converts the score to binary labels for classification; **CED** [29], exploiting specially designed auxiliary and oracle samples to enhance the distinction between in-distribution and out-of-distribution data; and **LLM-Check** [45], employing token logits and eigenvalue analysis of internal representation for detection, from which we choose the best-performing hidden state (LLM-Check-h) and attention-based variations (LLM-Check-a). All these methods are based on the last generated token as settled in their original paper. Notably, SAPLMA utilises a four-layer multilayer perception (MLP) with ReLU non-linearity while the other methods, including HaMI, employ a detector with two layers. We also assess the performance of HaMI using varying number of layers in Section 5.3.

Different variants of HaMI are evaluated in our experiments depending on the instantiation of Eq. 8. **HaMI** uses $P^c_{\text{uncertainty}}$ in Eq. 8 by default due to its superior performance, while **HaMI**$^*$ is the basic variant of HaMI that does not involve Eq. 8, using solely the original features. $\lambda$ in Eq. 8 is set to 1.0 by default. Unless otherwise specified, the hidden dimension of the two-layer MLP is set to 256. $k$ in Eq. 1 is dynamically determined by the generated token length ($l$), defined as $k = \lfloor 0.1 \times l \rfloor + 1$.

**Evaluation.** Following [17, 42], we evaluate the model's capability for hallucination detection by calculating the area under the receiver operating characteristic curve (AUROC). The ground-truth labels indicating the correctness are given by GPT-4.1 [1], which is instructed to determine if the answer is correct or not based on the consistency between generated answers and gold answers and its own knowledge. The prompt template follows [17] (see more details in Appendix A.2). Note that since GPT-4.1 makes mistakes mostly for the positive samples, we ask GPT-4.1 to rejudge samples labelled as positive and discard samples if the result is inconsistent with the first result. For this reason, the final sizes of the training and testing sets vary around $1,900$ and $400$, respectively. Additionally, we also conduct experiments where answers are evaluated with BLEURT score [43]. Results are presented in Appendix D.

## 5.1 Main Results

In Table 1, we evaluate our proposed HaMI by comparing it with eight SotA competing detection methods on four diverse QA datasets in LLMs from two different families of various sizes. As depicted in the table, our approach achieves superior performance compared with other methods in all three LLMs. In particular, HaMI$^*$ consistently and substantially outperforms all methods that do not require assistance from external LLMs, including Perplexity, CCS, SAPLMA, HaloScope, CED, and LLM-Check. When compared to approaches that leverage LLMs at the post-generation stage, HaMI$^*$

| | LLaMA-3.1-8B | | | | Mistral-Nemo-Instruct (12B) | | | | LLaMA-3.3-Instruct-70B | | | |
|---|---|---|---|---|---|---|---|---|---|---|---|---|
| | Trivia QA | SQuAD | NQ | BioASQ | Trivia QA | SQuAD | NQ | BioASQ | Trivia QA | SQuAD | NQ | BioASQ |
| p(True) [25] | 0.666 | 0.614 | 0.650 | 0.673 | 0.862 | 0.749 | 0.772 | 0.749 | 0.563 | 0.596 | 0.530 | 0.573 |
| Perplexity [42] | 0.732 | 0.649 | 0.659 | 0.709 | 0.720 | 0.662 | 0.646 | 0.675 | 0.671 | 0.626 | 0.619 | 0.583 |
| SE [17] | 0.828 | **0.787** | 0.773 | 0.757 | 0.795 | 0.752 | 0.813 | 0.800 | 0.819 | 0.643 | 0.769 | 0.772 |
| MARS [6] | 0.766 | 0.663 | 0.661 | 0.706 | 0.765 | 0.693 | 0.700 | 0.710 | 0.704 | 0.648 | 0.635 | 0.610 |
| MARS-SE [6] | 0.824 | 0.780 | 0.777 | 0.744 | 0.797 | 0.743 | 0.786 | 0.798 | 0.794 | 0.676 | 0.772 | 0.769 |
| CCS [9] | 0.675 | 0.596 | 0.628 | 0.662 | 0.551 | 0.579 | 0.615 | 0.553 | 0.597 | 0.575 | 0.592 | 0.562 |
| SAPLMA [5] | 0.835 | 0.769 | 0.781 | 0.821 | 0.860 | 0.797 | 0.836 | 0.859 | 0.842 | 0.672 | 0.817 | 0.748 |
| HaloScope [14] | 0.610 | 0.669 | 0.637 | 0.635 | 0.653 | 0.621 | 0.568 | 0.676 | 0.656 | 0.655 | 0.622 | 0.556 |
| CED [29] | 0.745 | 0.776 | 0.710 | 0.695 | 0.693 | 0.680 | 0.669 | 0.674 | 0.709 | 0.699 | 0.714 | 0.703 |
| LLM-Check-h [45] | 0.666 | 0.614 | 0.610 | 0.673 | 0.696 | 0.659 | 0.673 | 0.668 | 0.678 | 0.639 | 0.642 | 0.702 |
| LLM-Check-a [45] | 0.651 | 0.644 | 0.626 | 0.675 | 0.715 | 0.677 | 0.690 | 0.670 | 0.691 | 0.657 | 0.653 | 0.713 |
| HaMI* (Ours) | **0.854** | 0.783 | **0.788** | **0.823** | **0.883** | **0.813** | **0.843** | **0.873** | **0.858** | **0.765** | **0.820** | **0.813** |
| HaMI (Ours) | **0.897** | **0.826** | **0.820** | **0.836** | **0.903** | **0.837** | **0.867** | **0.888** | **0.891** | **0.774** | **0.846** | **0.825** |

Table 1: AUROC results of HaMI and its competing methods on four datasets with three LLMs. HaMI* denotes a basic HaMI using the original representations as inputs. The best results are in red and second-best are in blue.

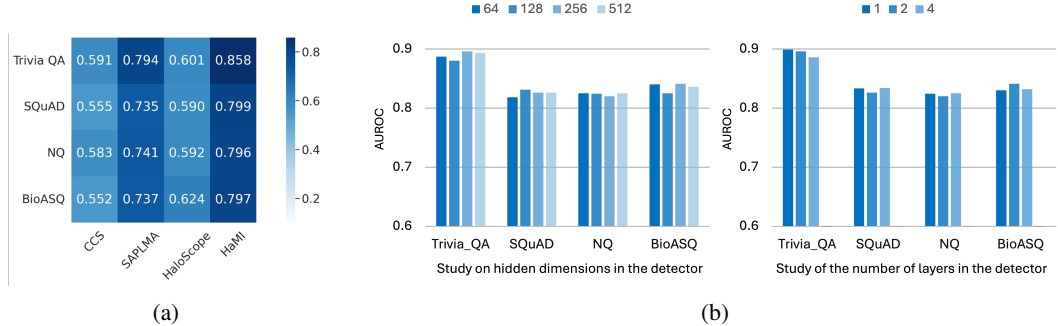

Figure 3: **(a)** AUROC results of cross-dataset generalisation on four datasets using LLaMA-3.1-8B. **(b)** AUROC w.r.t. dimensionality of the feature layer (**left**) and the number of network layers (**right**) based on LLaMA-3.1-8B.

achieves similar superiority across all cases except on the SQuAD dataset with LLaMA-3.1-8B, where its performance is comparable to the best score achieved by the SE method.

Moreover, with the augmentation of the uncertainty measurement, we observe that HaMI significantly outperforms all SotA methods by a large margin. It gains as large as 8.1% to 11.9% averaged AUROC improvement over MARS-SE, which also leverages entailment information among various generations, in three LLMs. Notably, p(True), MARS and Semantic Entropy all resort to external LLMs for assistance, but their capability for hallucination detection differs significantly. Following the configurations described in their respective original studies, p(True) and MARS employ relatively smaller and less capable models, whereas Semantic Entropy harnesses the more powerful GPT-3.5 model. This variation underscores that the performance of uncertainty-based methods requiring multiple sampling is highly dependent upon the capabilities of the selected assistant LLM.

Furthermore, among methods based on internal representations, supervised approaches generally outperform the semi-supervised/unsupervised ones. For example, SAPLMA and our approach exhibit superior performance to CCS and HaloScope. Notably, both HaMI* and HaMI outperform SAPLMA and demonstrate consistently strong results across all four benchmarks. This superiority is due to not only the predictive uncertainty enhancement but also the effectiveness of adaptive token selection on generation responses of diverse forms (see Section 5.3 for more detailed analysis).

Comparing the performance on LLaMA models with various scales, we observe that HaMI achieves a notably larger margin of improvement compared to baselines for larger LLaMA-3.3-Instruct-70B. For instance, on the LLaMA-3.1-8B model, HaMI surpasses SE and SAPLMA by an average of 7.4% and 5.5%, respectively. On the larger model, these improvements increase to 11.5% and 8.7%, representing a relative improvement of over 50%.

| | LLaMA-3.1-8B | | Mistral | |
|---|---|---|---|---|
| | Trivia QA | SQuAD | Trivia QA | SQuAD |
| First | 0.849 | 0.774 | 0.844 | 0.799 |
| Before Last | 0.878 | 0.778 | 0.849 | 0.807 |
| Last | 0.890 | 0.804 | 0.873 | 0.825 |
| **Ours** | **0.897** | **0.826** | **0.903** | **0.837** |

| | LLaMA-3.1-8B | | Mistral | |
|---|---|---|---|---|
| | Trivia QA | SQuAD | Trivia QA | SQuAD |
| Original | 0.854 | 0.783 | 0.883 | 0.813 |
| $P^t_{uncertainty}$ | 0.856 | 0.782 | 0.884 | 0.825 |
| $P^s_{uncertainty}$ | 0.871 | 0.787 | 0.897 | 0.831 |
| $P^c_{uncertainty}$ | **0.897** | **0.826** | **0.903** | **0.837** |

Table 3: Ablation study results on the ATS module (**left**) and uncertainty-enabled representation module (**right**).

## 5.2 Cross-dataset Generalisation Ability

The ability to generalise across datasets is essential to facilitating real-world applications of LLMs in diverse domains. We conduct experiments on the aforementioned four datasets to assess whether the proposed HaMI can effectively generalise among various datasets, with three strong competing methods in the line of internal-representation analysis as baselines. For each dataset, we report the average AUROC scores of detectors trained on one of the other three datasets. The results are illustrated in Figure 3a. It is clear that our method HaMI consistently achieves the best generalisation performance across all four datasets, outperforming the best competing method SAPLMA by 7% - 9%. Compared to the within-dataset performance in Table 1, the maximum performance decline in HaMI is no more than $4.5\%$, observed on the BioASQ dataset, which is significantly lower than that of the competing methods. We also compare the generalisation capability of HaMI with training-free baselines. The results for HaMI trained on one dataset and evaluated on others separately are presented in Table 2, where the results for the training-free baselines are the best AUROC of training-free methods—p(True), Perplexity, SE, MARS, MARS-SE, CED, LLM-Check-h, and LLM-Check-a—on each dataset. Notably, across all settings, HaMI can consistently outperform training-free/non-learnable baselines on unseen target datasets, regardless of which dataset it is trained on. The observation is similar to the one with the results aggregated over multiple detectors. These findings affirm the effectiveness of HaMI as a reliable hallucination detector in unseen datasets.

| | Trivia QA | SQuAD | NQ | BioASQ |
|---|---|---|---|---|
| **Trained on Trivia QA** | - | 0.802 | 0.818 | 0.796 |
| **Trained on SQuAD** | 0.850 | - | 0.794 | 0.804 |
| **Trained on NQ** | 0.870 | 0.804 | - | 0.791 |
| **Trained on BioASQ** | 0.853 | 0.790 | 0.781 | - |
| **Training-Free Baselines** | 0.828 | 0.787 | 0.777 | 0.757 |

Table 2: Cross-dataset generalization capability compared with training-free baselines.

## 5.3 Ablation Study

**Analysis of MIL-based Adaptive Token Selection.** Unlike existing methods that predetermined critical tokens for capturing truthfulness information, HaMI proposes the MIL-based adaptive token selection (ATS) module. Table 3 **Left** compares the performance of our ATS module with commonly used methods, including the *First* generated token, the *Last* generated token, and the *Before Last* generated token. The results show that our ATS method outperforms the alternatives across both Trivia QA and SQuAD datasets on both LLMs, yielding an average improvement of 6% over the *First* token, 4% over the *Before Last* token and 2% over the *Last* token.

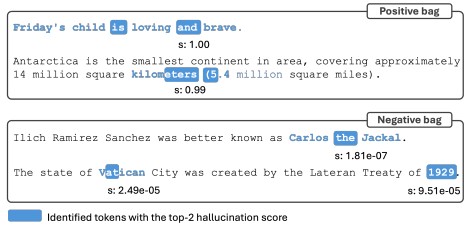

Figure 4: Adaptive token selection results showing tokens with the top-2 highest hallucination scores.

Furthermore, our ATS module exhibits substantially better robustness to the length of generations than the competing methods. In particular, the average number of generated tokens for the Trivia QA dataset with the LLaMA model is 10, which is smaller than the other three cases (Trivia QA with Mistral model - 13.5, SQuAD with LLaMA model - 17.5, SQuAD with Mistral model - 15). Despite this variation, the ATS module performs robustly on all four cases and even performs more effectively as the length increases. In contrast, the performance of the predefined token location methods is unstable and tends to degrade as the length of the generation grows. Moreover, the *Last* token achieves better performance across all scenarios, and the *First* token is just the opposite. It

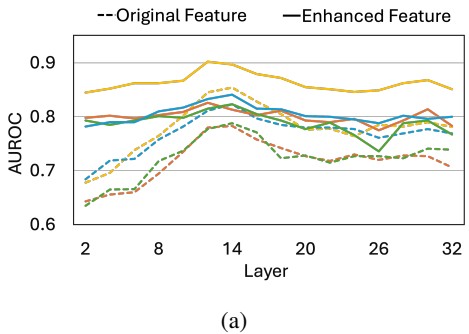 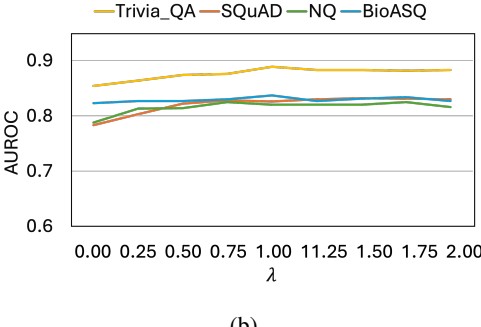

(a)                                                              (b)

Figure 5: **(a)** Performance of HaMI w.r.t. internal representations extracted from different layers. **(b)** The impact of augmentation strength $\lambda$ in Eq. 8 on HaMI. All results are based on LLaMA-3.1-8B.

suggests that the last token can capture full semantics under specific length conditions, while the first token may miss content in the subsequent predictions.

To further support the validity of the token selection process, we conducted an additional study, a manual evaluation using 100 randomly sampled examples from the Trivia QA dataset. In this analysis, we manually annotated hallucinated tokens and measured the token detection recall rate of our method throughout training. Specifically, HaMI can achieve a recall rate at 0.84 as training progresses. This suggests that our approach enables the model to identify salient tokens even without token-level supervision. Figure 4 presents an illustrative example of the scoring results over tokens in a positive bag and a negative bag, where we can observe that there is significant distinguishability between the positive and negative tokens since the maximum scores of the instances in the positive bag are substantially greater than those in the negative bag. Tokens denoted by blue characters have scores that are very close to the largest ones and we find that these tokens can be adaptively identified by our method as concentrated answers in comparison to ground-truth answers. Additionally, it is noted that while selected tokens are associated with the exact answer, they may appear at any location in their vicinity, which can be captured by our smoothness loss.

**Analysis of Different Predictive Uncertainty-enabled Features.** Here we systematically examine three uncertainty measuring methods presented in Section 4.2 for the internal representation enhancement as detailed in Eq. 5, 6 and 7 respectively. We use a simple combination strategy as defined in Eq. 8. In the right panel of Table 3, we can observe that both $P^s_{\text{uncertainty}}$ and $P^c_{\text{uncertainty}}$ contribute to improved detection capabilities over the baseline (*i.e.,* the original feature representations) while the token-level uncertainty $P^t_{\text{uncertainty}}$ yields limited effectiveness. Specifically, the enhancements attributed to semantic consistency $P^c_{\text{uncertainty}}$ are the most significant, exhibiting improvements up to 6.7%. These observations suggest that the effectiveness of uncertainty-based enhancement is tied to the inherent ability of the uncertainty metric to distinguish hallucinations from correct responses. Notably, although the improvements observed with $P^s_{\text{uncertainty}}$ are less pronounced, it surpasses the performance of SotA multi-sampling approaches (such as SE in Table 1), without incurring the costs associated with multiple generations and external LLM employments. This highlights the potential of HaMI for deployments in various practical environments that involve external tools or not.

**Performance of HaMI w.r.t. Representations from Different LLM Layers.** We evaluate detection performance using representations extracted from various layers of LLM with LLaMA-3.1-8B across all benchmarks. We apply both original internal representations and semantic-equivalence-enhanced representations for investigation. As depicted by the dashed line in Figure 5a, the AUROC values for original representations exhibit a clear increase, peaking between layers 12 and 16, before declining to a relatively stable level. This observation suggests that the truthfulness content evolves across the initial to middle layers. The performance of the uncertainty-enhanced representations (presented in the solid line) maintains a relatively consistent trend, with the highest AUROC scores concentrated in the middle layers as well. The comparison of these trends indicates that incorporating predictive uncertainty enhances the distinctiveness of the representations, particularly in the earlier layers where less semantic information is typically available.

**Performance of HaMI w.r.t. Various Model Scales.** We investigate the sensitivity of HaMI on the model scale by varying the number of layers and hidden dimensions in the two-layer MLP network. As illustrated in Figure 3b, HaMI exhibits low sensitivity to both hyperparameters across all four benchmarks. Specifically, for a two-layer MLP (employed in HaMI and most baselines), the performance remains steady as the size of the hidden dimension changes from 64 to 512. Similarly, increasing the number of layers from 1 to 4 yields only marginal changes in AUROC, suggesting that a shallow architecture is sufficient for effective hallucination detection. These findings highlight the robustness of our method *w.r.t.* variations in network architecture.

**Performance of HaMI w.r.t. Uncertainty Augmentation Strength ($\lambda$).** Figure 5b presents results based on various choice of $\lambda$ used in Eq. 8. It is clear that the model's performance is steadily improved as $\lambda$ changed from 0.0 towards 1.0. Beyond this point, performance becomes stable.

## 6   Conclusion

In this paper, we introduce the very first approach that supports joint token selection and hallucination detection, HaMI, enabling adaptive identification of the most hallucinated tokens for learning an optimal hallucination detector. This helps largely enhance the robustness of the detection on generation responses of varying lengths and hallucinated entities. Specifically, HaMI incorporates a straightforward yet effective MIL formulation to automatically highlight salient tokens that are optimal for training the subsequent hallucination detector. Additionally, we also explore integrating uncertainty metrics into the original representations to enrich them with more information about truthfulness. Extensive empirical results demonstrate that HaMI substantially outperforms existing SotA methods across diverse popular datasets and LLMs. Our ablation studies offer important insights into how different designs in HaMI help improve the robustness of the detection. While our experiments primarily focus on the QA domain, the principles underlying our method are task-free, suggesting potential applicability to a broad spectrum of other tasks.

## Acknowledgments

This research is supported by A*STAR under its MTC YIRG Grant (M24N8c0103), the Ministry of Education, Singapore under its Tier-1 Academic Research Fund (24-SIS-SMU-008), and the Lee Kong Chian Fellowship.

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

# A Implementation Details

## A.1 Setup

For the main results, our hallucination detector $f_\theta(\cdot)$ utilises a two-layer MLP with a hidden dimension of 256. The first linear layer is followed by a BatchNorm and ReLU activations and the second linear layer is accompanied by a sigmoid output. As shown in Figure 3b in our paper, our method is relatively insensitive to variations in the number of layers and the hidden dimension size. At the training stage, we use Adam optimiser. For the input representations of our detector, we utilise representations extracted from a single layer and the layer index is determined by the results of the validation set.

We implement our method using PyTorch 2.6.0 [39] and transformers 4.51.3 [48] and conduct all experiments on NVIDIA A100 GPUs. Based on experimental records, the estimated total training and inference time for each dataset under the main experimental settings is as follows: 0.40 GPU-hours for LLaMA-3.1-8B, 0.45 GPU-hours for Mistral-Nemo-Instruct (12B) and 0.65 GPU-hours for LLaMA-3.3-Instruct-70B with 8-bit quantisation. It is important to note that all these results are based on the assumption that the semantic uncertainty score $P^c_{\text{uncertainty}}$ is readily available. Otherwise, computing this score involves generating multiple responses and evaluating entailment across them, which introduces nontrivial latency, approximately 7.6 seconds per question for LLaMA-3.1-8B, 8.6 seconds for Mistral-Nemo-Instruct (12B), and 46.1 seconds for LLaMA-3.3-Instruct-70B. In this work, we also propose a lightweight variant, HaMI*, using only a single generation, to eliminate the influence on computational efficiency while maintaining superior performance.

## A.2 Prompts for Generation and Evaluation

All prompts utilised in our experiments refer to those used by Farquhar *et al.* [17]. For the QA tasks, we prompt selected LLaMA and Mistral models to generate answers without context in a zero-shot manner. Figure 6 presents the used input prompt. To obtain response labels, we prompt GPT-4.1 [1] with OpenAI API to evaluate the quality of the response. The prompts are illustrated in Figure 7. It is observed that GPT-4.1 can make mistakes for the positive samples as checked with the gold answers. Therefore, we ask GPT-4.1 to rejudge samples labelled as positive and discard samples if the result is inconsistent with the first result.

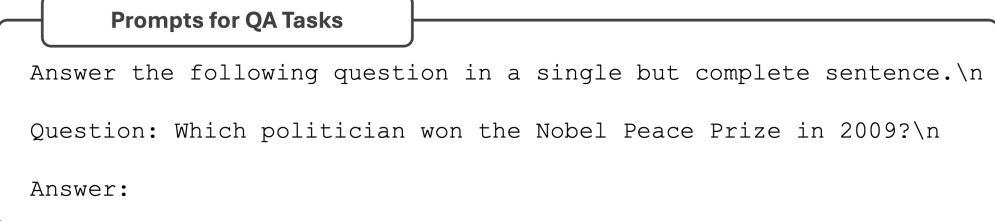

Figure 6: Prompts for QA tasks without context.

# B Study on the Effectiveness of the Smoothness Loss

The smoothness loss is designed under the assumption of continuity in hallucination scores between adjacent tokens. We investigate the effectiveness of smoothness loss on four benchmarks with LLaMA-3.1-8B model. Main results on each individual dataset and cross-dataset results are respectively presented in Table 4 and Table 5 below. The results are averaged over three independent runs. As evidenced in the tables, incorporating the smoothness loss consistently yields improved performance across all the cases. On average, the smoothness loss consistently enhances HaMI, increasing the performance by 0.7% and 1.1% on the within-dataset and cross-dataset scenarios, respectively. This demonstrates that the contribution of the smoothness loss is more pronounced in the more challenging cross-dataset setting. These consistent improvements highlight that the smoothness loss plays a rather positive role in helping achieve a more robust and generalizable HaMI, which is particularly valuable in cross-domain scenarios.

```
We are assessing the quality of answers to the following question:
{Qusetion} \n

The expected answer is: {Expected_Answer}.\n
- or: The following are expected answers to this question:
{Expected Answers}.\n

The proposed answer is: {predicted_answer}\n

Within the context of the question and your own knowledge, is the
proposed answer correct or the same as the expected answer?
Respond only with yes or no.\nResponse:
```

Figure 7: GPT-4.1 evaluation prompts

| $\mathcal{L}_{MIL}$ | $\mathcal{L}_{smooth}$ | Trivia QA | SQuAD | NQ | BioASQ |
|---|---|---|---|---|---|
| ✓ | | 0.892±0.005 | 0.825±0.002 | 0.812±0.003 | 0.840±0.015 |
| ✓ | ✓ | 0.898±0.007 | 0.835±0.006 | 0.815±0.004 | 0.845±0.012 |

Table 4: Ablation study on the MIL and smoothness losses.

| $\mathcal{L}_{MIL}$ | $\mathcal{L}_{smooth}$ | Trivia QA | SQuAD | NQ | BioASQ |
|---|---|---|---|---|---|
| ✓ | | 0.852±0.012 | 0.812±0.005 | 0.797±0.020 | 0.790±0.007 |
| ✓ | ✓ | 0.865±0.008 | 0.820±0.002 | 0.808±0.012 | 0.795±0.002 |

Table 5: Ablation study on the MIL and smooth losses under the cross-dataset setting.

## C  Study on Top-$k$ Token Selection

HaMI employs a hard top-$k$ selection strategy in its MIL formulation that functions similarly to max pooling during training. This design aligns with the core assumption in multi-instance learning (MIL) that if there are any positive instances, the bag should be labelled as positive. This principle is particularly relevant to our detection task, where the salient hallucinated tokens appear sparsely within generated sequences. While there exist more sophisticated and learnable aggregation techniques, previous research has indicated that, given a sufficient number of training bags, different aggregation methods for a bag tend to converge to similar performance [20]. Based on these observations and the remarkable performance of our method in our experimental results, despite being simple, the hard top-$k$ strategy offers a direct, yet effective solution for the detection task. As presented in Section 4.1, $k$ is dynamically determined by the generated token length $l$ as follows:

$$k = \lfloor r_k \times l \rfloor + 1, \tag{9}$$

where $r_k$ is set to 0.1 by default. In this section, we investigate the sensitivity of our method to the choice of $r_k$ in the adaptive token selection process. As shown in Figure 8, HaMI exhibits stable performance across a set of $r_k$, ranging from zero to 0.20, on all four datasets with LLaMA-3.1-8B and Mistral-Nemo-Instruct (12B). HaMI can perform stably across different choices of $r_k$. Notably, smaller $r_k$ values already yield strong results, which implies that only a few high-salience tokens are sufficient for effective hallucination discrimination.

## D  Evaluation results with BLEURT score

In addition to labels identified by GPT-4.1 [1], we also use the BLEURT-threshold [43] method to generate labels and comparing HaMI with five representative baselines. Results are reported in Table 6, which shows that our methods HaMI and HaMI* can also consistently outperform the

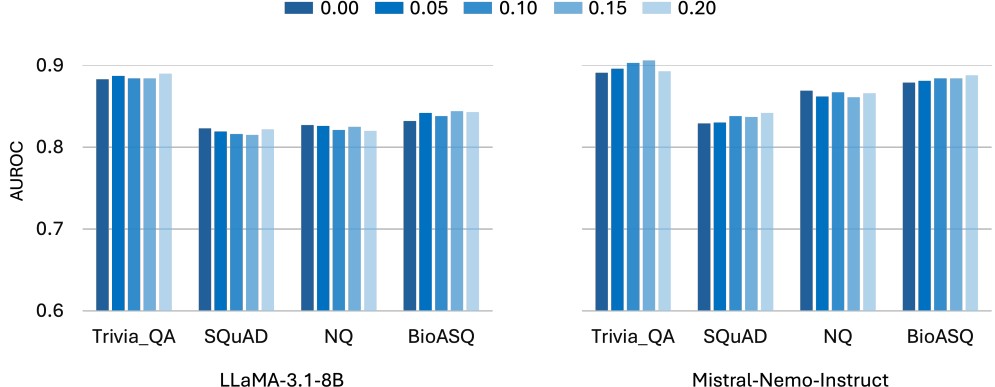

Figure 8: Study on different choices of $r_k$ across four datasets with two LLMs.

| Method | Trivia QA | SQuAD | NQ | BioASQ |
|--------|-----------|-------|-----|--------|
| Perplexity | 0.663 | 0.638 | 0.586 | 0.662 |
| SE | 0.709 | 0.813 | 0.704 | 0.714 |
| CCS | 0.682 | 0.657 | 0.738 | 0.683 |
| SAPLMA | 0.897 | 0.898 | 0.867 | 0.873 |
| HaloScope | 0.792 | 0.845 | 0.852 | 0.830 |
| HaMI* | 0.900 | 0.928 | 0.909 | 0.918 |
| **HaMI** | **0.909** | **0.937** | **0.920** | **0.922** |

Table 6: AUROC results with labels generated by the BLEURT method. The results are based on LLaMA-3.1-8B.

baselines using the BLEURT-based labelled datasets. These results provide empirical evidence for the robustness of HaMI from another perspective (*i.e.,* a data labelling perspective).

## E    Limitations and Broader Impacts

**Broader Impacts.**    With the growing deployment of LLMs, ensuring the truthfulness of their outputs has become a critical challenge, especially in high-stakes domains such as finance and healthcare. To enable effective hallucination detection, in this work, we propose a practical and robust approach that leverages internal representations to identify tokens with a high likelihood of being hallucinated and enables the following detection. We hope that our proposed method, HaMI, can contribute to safer and more reliable real-world deployment of LLMs. Although our research focuses on QA tasks, HaMI can be extended to other tasks as well. For example, in the summarisation task, truthfulness can be assessed sentence by sentence. Verifying the correctness of each generated sentence can be reformulated as an entailment query against the source context, allowing hallucination detection to be performed in a QA-like manner. Furthermore, the similarity between a generated sentence and its corresponding context could serve as an uncertainty metric for internal representation enhancement.

**Limitations.**    Apart from the aforementioned capabilities of HaMI, there remain some concerns. Unlike some black-box uncertainty-based approaches, our method requires access to internal representations, limiting its deployments to open-source LLMs. Moreover, we also explore integrating uncertainty metrics into the original representations to enhance their discriminative capability on correct and incorrect generations. While effective, the integration strategy is relatively simple and we acknowledge that more sophisticated fusion methods could be explored for better detection performance in future research.

