# OpenReview forum: "Robust Hallucination Detection in LLMs via Adaptive Token Selection"
_NeurIPS.cc/2025/Conference — NeurIPS 2025 poster_

### Official Review · Reviewer_Z1J4 · 2025-06-21

**Clarity:** 3
**Significance:** 2
**Originality:** 3
**Rating:** 4
**Confidence:** 3

**Summary:**

The paper proposes HaMI, a hallucination detection framework for LLMs that formulates the task as a multiple instance learning (MIL) problem. Rather than relying on predetermined token positions (e.g., last token), HaMI adaptively selects salient tokens based on their hallucination scores and trains a detector end-to-end. The authors further enhance detection performance by incorporating predictive uncertainty into internal token representations. Extensive experiments across four QA benchmarks and multiple LLMs demonstrate strong improvements over state-of-the-art (SotA) baselines, particularly in cross-dataset generalization.

**Questions:**

1. The authors state that the model selects the top-k tokens based on hallucination scores. However, this operation is not differentiable. Do you simply treat top-k as a hard selection during training, or is there a differentiable approximation (e.g., using soft top-k, Gumbel-softmax, or similar techniques)? If it’s a hard selection, how is this really learning to select the right tokens? I would appreciate a clarification on that.
2. In lines 275–276 You mention, "For each dataset, we report the average AUROC scores of detectors trained on one of the other three datasets."
     - Does this mean you trained three detectors, each on a different dataset and then averaged their performance on the target dataset?
     - Or was only one source dataset chosen (randomly or fixed) for training, and results reported per target? If so, how did you choose it?
 Clarifying this setup is important to interpret generalization performance.
3. In Figure 3(a), cross-dataset generalization is only meaningful if it outperforms non-learnable baselines, such as those based on token probabilities or logits. To strengthen their results, I suggest the authors provide a justification for why such probability/logit-based baselines would perform worse than their method. Ideally, and if time permits, an additional experiment could be included to compare against a simple heuristic—such as averaging the token probabilities or logits across a sentence—as a baseline for hallucination detection in the cross-dataset setup.
4. In Table 2 (left), the performance gap between “ours” and probing the last token appears relatively small. While this might be acceptable, it’s important to note that “ours” includes additional information from predictive uncertainty. At this point, it’s unclear whether the observed improvement stems from the use of predictive uncertainty or from the top-k token selection itself. Clarification on this would be helpful. If feasible, I would also recommend adding a new row to Table 2 (left) showing results for a variant of HaMI without predictive uncertainty (e.g., HaMI*), to isolate the contribution of top-k selection.

**Ethical Concerns:**

["NO or VERY MINOR ethics concerns only"]

**Final Justification:**

I think this is a solid, incremental contribution. I would be happy to see it accepted, but I would also understand a rejection, given the areas for improvement outlined. As such, I will maintain my score as a borderline accept.

**Limitations:**

The authors acknowledge that their method is limited to open-source LLMs due to its reliance on internal representations. They also note that their integration of uncertainty metrics is simple and could be improved with more advanced fusion techniques in future work.

**Paper Formatting Concerns:**

I don’t have any formatting concerns.

**Quality:**

2

**Strengths And Weaknesses:**

**Strengths:**
- The paper is well-written and easy to follow.
- Recasting hallucination detection as a MIL problem is original and well-motivated.
- The authors compare against both internal representation and uncertainty-based methods across a broad suite of datasets and models and show superior performance.


**Weaknesses:**
- The central claim of this work is "adaptive token selection," but if I understand correctly, this adaptation is not actually learned. The method selects the top-k candidates, relying on the hope that the model will implicitly learn to choose the appropriate tokens during training. However, top-k selection is a non-differentiable operation, making it analogous to the max operation in convolutional layers—where backpropagation does not learn which input to pick, but merely propagates gradients through the selected maximum. This raises a concern: it's unclear how much the proposed "adaptive token" mechanism truly contributes to performance, especially when compared to simply probing the last token (see Question 4 below).
- Minor concern: The proposed method appears to ignore the sequential nature of the tokens. If I understand correctly, it simply averages over the top-k candidates without considering their order or relative positions—information that could be valuable for detection tasks.

---

> ### Author Rebuttal · Authors · 2025-07-31
>
> Thank you very much for the constructive comments. We are grateful for the positive comments on our innovation and empirical justification. Please see our response to your comments one by one below.
>
> > **Question #1 \& Weakness #1**: Clarification on the top-k tokens based on hallucination scores. If it’s a hard selection, how is this really learning to select the right tokens?
>
> We confirm that our current implementation employs a hard top-k selection strategy during training, without using a differentiable approximation. Although the selection operation itself is non-differentiable, the learning process remains effective due to the design of our Multiple Instance Learning (MIL) framework, where instance selection is treated as a latent and optimization is performed over the selected subset. Specifically, while the top-k selection is discrete, the hallucination scores for each token are generated by a fully differentiable detector network. Gradients from the MIL loss are backpropagated through this scoring function, allowing the model to learn how to assign more accurate scores with better supervision over time.
>
> To validate the effectiveness of our approach, we manually annotated hallucinated tokens in 100 randomly selected samples and conducted a token-level recall analysis on the TriviaQA dataset. Our adaptive token selection achieved a recall rate of 0.84, indicating that the model is indeed learning to select tokens highly aligned with the exact hallucinations. This recall performance is remarkable considering that we do not have token-level hallucination labels.
>
> Please also kindly refer to our responses to Reviewer **NwzN**'s **Question #1**, **Weakness #4**, and Reviewer **dBF6**'s **Question #2 \& Weakness #3** for more discussion on addressing this concern.
>
> > **Question #2**: Clarification on "For each dataset, we report the average AUROC scores of detectors trained on one of the other three datasets." (lines 275–276)
> > - Does this mean you trained three detectors, each on a different dataset and then averaged their performance on the target dataset?
> > - Or was only one source dataset chosen (randomly or fixed) for training, and results reported per target? If so, how did you choose it? Clarifying this setup is important to interpret generalization performance.
>
>
> As correctly noted in your first point, given the four datasets, our cross-dataset evaluation follows the setup where detectors are trained separately on each of the three source datasets (leaving the target dataset out, one detector per source dataset), and we then evaluate the detectors on the target dataset and report the AUROC score averaged over the results of the three detectors. This setting is designed to assess the model's ability to generalize from various source datasets to the target dataset. We appreciate your attention to this part of experimental design, and we will clarify this point in the paper.
>
>
> > **Question #3**: In Figure 3(a), cross-dataset generalization is only meaningful if it outperforms non-learnable baselines, such as those based on token probabilities or logits. To strengthen their results, I suggest the authors provide a justification for why such probability/logit-based baselines would perform worse than their method. Ideally, and if time permits, an additional experiment could be included to compare against a simple heuristic—such as averaging the token probabilities or logits across a sentence—as a baseline for hallucination detection in the cross-dataset setup.
>
> Thank you very much for the question. The predictive logits of the tokens reflect how likely a token is to be generated by an LLM, which is related to training distribution of the model. As such, probability- or logit-based measurements would be influenced by fluency and grammaticality while hallucinations often occur in fluent but factually incorrect text. This is often one main cause to their degraded performance in hallucination detection. In contrast, internal representations extracted from LLMs can encode deeper semantic and contextual information, including how the model encodes, processes and grounds knowledge, which allows for a more nuanced assessment of whether the model’s output is correct or not. These observations align with findings in previous studies [Ref1], which suggest that “LLMs know more than they express”.
>
> For the comparison of cross-dataset generalization capability, we selected CCS, SAPLMA, and HaloScope because these methods access the internal states of LLMs, which is also applied in HaMI. We agree that including non-learnable/training-free baselines is also important for a comprehensive assessment of a method’s generalization ability. For the non-learnable methods, including p(True), Perplexity, SE, and MARS, their results in Table 1 reflect their cross-dataset performance since they involve no training on source datasets.
> Comparing these results to the results of HaMI in Figure 3 (a), it is clear that HaMI consistently outperforms these state-of-the-art non-learnable baselines. Specifically, the cross-dataset performance of HaMI on four datasets are **0.858, 0.799, 0.796, 0.797**, which is higher than the best results on each individual dataset, **0.824, 0.787, 0.777, 0.757**, that can be obtained by these non-learnable baselines (these results are based on single running of HaMI; the same conclusion can be drawn for the averaged performance of HaMI over three independent runs in **Table A3** in our response to Reviewer **NwzN**'s **Question #3** & **Weakness #2**). Furthermore, the suggested simple logit baseline that averages token log-probabilities across the sentence achieves AUROC scores of only **0.732, 0.649, 0.659, 0.709** on the four datasets, which are significantly lower than the corresponding AUROC scores from HaMI.
>
> We will add this additional justification and comparison in the revised version of the paper to address this concern.
>
> > **Question #4**: In Table 2 (left), the performance gap between “ours” and probing the last token appears relatively small. While this might be acceptable, it’s important to note that “ours” includes additional information from predictive uncertainty. At this point, it’s unclear whether the observed improvement stems from the use of predictive uncertainty or from the top-k token selection itself. Clarification on this would be helpful. If feasible, I would also recommend adding a new row to Table 2 (left) showing results for a variant of HaMI without predictive uncertainty (e.g., HaMI*), to isolate the contribution of top-k selection.
>
> We would like to clarify that all the four variants of the method reported in Table 2 (left), including the “Last” and "Ours", are built upon the features enhanced with the uncertainty measure $P^{c}_{\text{uncertainty}}$, i.e., the four methods are all variants of HaMI rather than HaMI*  in Table 1 in our paper.  Therefore, the performance improvements observed can be solely attributed to the effectiveness of our token selection strategy rather than the uncertainty modelling.
>
> Additionally, please also refer to the results of Table 1, where we compare HaMI with HaMI* and SAPLMA using three different LLM models. HaMI is the model using both uncertainty modelling and adaptive token selection, HaMI* is trained directly on the raw features extracted from LLMs without using uncertainty modelling but using adaptive token selection, while SAPLMA relies solely on the last-token representations and does not use both uncertainty modelling and adaptive token selection. The differences between HaMI* and SAPLMA solely highlight the contribution of our adaptive token selection, while the differences between HaMI and HaMI* solely highlight the contribution of the uncertainty modelling.
>
>
> > **Weakness #2**: Minor concern: The proposed method appears to ignore the sequential nature of the tokens. If I understand correctly, it simply averages over the top-k candidates without considering their order or relative positions—information that could be valuable for detection tasks.
>
> Thank you for your insightful question.
> Our method indeed directly averages over the top‑k token candidates, without explicitly modelling their sequential order or relative positions. This design choice is based on the primary motivation of HaMI, aiming to learn which tokens are most informative for hallucination detection, regardless of their position in the generated sequence.
> By doing so, HaMI can robustly detect hallucination signals even when they appear sparsely or in varied locations across long generations. Therefore, the lack of explicit positional encoding in our MIL learning framework does not compromise performance, yet it helps our model adaptively select hallucination tokens in varying positions of the generated response. We agree that further incorporation of the sequence order may provide additional value; we will explore this addition under the MIL framework in our future work.
>
> **Reference:**
> - [Ref1] Azaria, Amos, and Tom Mitchell. The internal state of an LLM knows when it's lying. Findings of the Association for Computational Linguistics: EMNLP. 2023.

---

> > ### Comment · Area_Chair_jFGJ · 2025-08-06
> > **Please reengage**
> >
> > Dear reviewer Z1J4,
> >
> > This is a kind reminder that reviewers should participate in discussions with authors and submitting “Mandatory Acknowledgement”. Please try to engage with the authors based on their rebuttal before the deadline tonight.
> >
> > Thank you,

---

> > ### Comment · Reviewer_Z1J4 · 2025-08-06
> >
> > I thank the authors for the detailed response.
> >
> > First, the authors confirm that the selection mechanism is hard-coded rather than learned, which simplifies the overall methodology.
> >
> > Second, I thank you for clarifying the cross-dataset setup. While this configuration is somewhat indicative, it is not optimal, as it does not directly reflect a real-world deployment scenario. In practice, you're aggregating knowledge from multiple detectors, whereas the baselines are applied directly to the target dataset — a setup that arguably better simulates realistic conditions. Therefore, the comparison, while informative, is suboptimal.
> >
> > Regarding the baselines, as indicated by the authors response, the non-learnable methods evaluated by the authors include: P(True), Perplexity, SE, and MARS. However, this evaluation still omits some simple probability- or logits-based baselines, which have been shown to outperform for example P(True) in some cases (e.g., see Table 1 in [1]). That said, I believe this is a relatively minor concern.
> >
> > In conclusion, I think this is a modest contribution. I would be happy to see it accepted, but I would also understand a rejection, given the areas for improvement outlined above. As such, I will maintain my score as a borderline accept.
> >
> > References:
> >
> >  [1] LLMs Know More Than They Show: On the Intrinsic Representation of LLM Hallucinations. Orgad et al., ICLR 2025.

---

> > > ### Author Response · Authors · 2025-08-08
> > >
> > > We sincerely thank you for the valuable feedback. Below, we address each of your follow-up questions.
> > >
> > > > **1. Hard top-k selection strategy**
> > >
> > > We appreciate your observation regarding our token selection mechanism. Our top-k selection strategy functions similarly to max pooling. This design aligns with the core assumption in multi-instance learning (MIL) that if there are any positive instances, the bag should be labelled as positive. This principle is particularly relevant to our detection task, where the salient hallucinated tokens appear sparsely within generated sequences.
> > >
> > > While there exist more sophisticated and learnable aggregation techniques, previous research has indicated that given a sufficient number of training bags, different aggregation methods for a bag tend to converge to similar performance [Ref1]. Based on these observations and the remarkable performance of our method in our experimental results, despite being simple, the hard top-k strategy offers a direct, yet effective solution for the detection task. As the first work to formulate hallucination detection as a MIL problem, this solution provides novel insights into the relation of two sub-problems, token selection and hallucination scoring, opening up opportunities for more optimal solutions in this research line. We will clarify these points and discuss these future opportunities in the final version.
> > >
> > > > **2. On the cross-dataset setup and real-world realism**
> > >
> > > Regarding the concern about the limitations of our cross-dataset setup, we present the performance of detectors trained on one dataset and evaluated on others separately in Table A1 below.
> > >
> > > Notably, we observe that, across all settings, models trained on one dataset consistently outperform non-learnable baselines on unseen target datasets. The observation is similar to the one with the results aggregated over multiple detectors.
> > > This suggests a potential for HaMI for deployment in real-world settings where there is an auxiliary training dataset and training on the target domain is not feasible.
> > >
> > > |                             | Trivia QA | SQuAD | NQ   | BioASQ |
> > > |-----------------------------|-----------|--------|------|--------|
> > > | **Trained on Trivia QA**    | -         | 0.802  | 0.818| 0.796  |
> > > | **Trained on SQuAD**        | 0.850     | -      | 0.794| 0.804  |
> > > | **Trained on NQ**           | 0.870     | 0.804  | -    | 0.791  |
> > > | **Trained on BioASQ**       | 0.853     | 0.790  | 0.781| -      |
> > > | **Best AUROC for non-learnable methods** | 0.828 | 0.787  | 0.777| 0.757  |
> > >
> > > ```
> > > Table A1: Cross-dataset generalization capability compared with non-learnable/training-free baselines.
> > > ```
> > >
> > > > **3. Minor concern: on omitted simple probability- or logits-based non-learnable baselines**
> > >
> > > In our earlier experiments, we evaluated simple token-level logit-based uncertainty metrics. However, we found that using token-level logits performed slightly worse than sequence-level aggregated uncertainty signals (i.e., "logit-mean" in Table 1 in [Ref2]). As such, we reported results on the stronger sequence-level metrics in our paper.
> > > We appreciate your valuable suggestion and would like to report the AUROC scores of the “logit-min” metric, which achieves **0.728, 0.666, 0.632, 0.688** on Trivia QA, SQuAD, NQ, and BioASQ with LLaMA-3.1-8B model, respectively. We will add these results into the final version of our paper.
> > >
> > > Thank you very much again for your constructive and insightful feedback and for your positive recommendation of our paper.
> > >
> > >
> > > - [Ref1] Attention-based Deep Multiple Instance Learning. Ilse et al., ICML 2018.
> > > - [Ref2] LLMs Know More Than They Show: On the Intrinsic Representation of LLM Hallucinations. Orgad et al., ICLR 2025.

---

### Official Review · Reviewer_tguY · 2025-06-24

**Clarity:** 3
**Significance:** 3
**Originality:** 3
**Rating:** 4
**Confidence:** 4

**Summary:**

This paper proposes HaMI, a novel hallucination detection method for large language models (LLMs) based on Multiple Instance Learning (MIL). Unlike prior work that relies on preselected tokens for detection (e.g., last token), HaMI introduces an adaptive token selection mechanism that jointly optimizes token scoring and hallucination classification in an end-to-end fashion. Furthermore, the model integrates token-level and sentence-level uncertainty into internal representations to improve discrimination between hallucinated and truthful outputs. The authors demonstrate state-of-the-art performance across four QA benchmarks (TriviaQA, SQuAD, NQ, and BioASQ) and three LLMs (LLaMA and Mistral variants). Empirical results show superior robustness and generalization over prior hallucination detection methods.

**Questions:**

How accurate are GPT-4.1 labels in practice? Could you report inter-agreement between the two judgment passes? Have you considered incorporating human annotations or knowledge-based QA datasets (e.g., where gold answer is guaranteed factual)?

**Ethical Concerns:**

["NO or VERY MINOR ethics concerns only"]

**Final Justification:**

The authors have conducted comprehensive comparisons and analyses against a broad set of baseline methods, providing solid empirical support for their approach. Overall, we consider this work to be a valuable and well-executed contribution that merits acceptance.

**Limitations:**

See the weakness

**Paper Formatting Concerns:**

No significant formatting issues observed.

**Quality:**

2

**Strengths And Weaknesses:**

Strengths:

Timely and relevant problem: Hallucination in LLMs is a critical safety issue, and detection methods remain underdeveloped.

Conceptual novelty: Reformulating hallucination detection as a MIL problem with adaptive token selection is original and well-motivated.

Clarity: The paper is well-written and well-structured, with clear methodological explanations and visualizations.

Weaknesses:

• Limited baseline comparison: Important recent hallucination detection methods such as LLM-Check and CED (Comparing Embedding Differences) are not included. These would provide stronger empirical context and help position the contribution more rigorously.

[1] Sriramanan, Gaurang, et al. "Llm-check: Investigating detection of hallucinations in large language models." Advances in Neural Information Processing Systems 37 (2024): 34188-34216.

[2] Lee, Hakyung, et al. "CED: Comparing Embedding Differences for Detecting Out-of-Distribution and Hallucinated Text." Findings of the Association for Computational Linguistics: EMNLP 2024. 2024.

• Label reliability: The ground truth annotations for hallucination are solely based on GPT-4.1 self-judgment, with only a two-pass validation. While practical, this approach lacks guarantees of correctness. Incorporating datasets with verified human or knowledge-grounded hallucination annotations would strengthen the empirical claims.

• Implicit token selection: The MIL approach does not guarantee that selected high-scoring tokens are truly the ones responsible for hallucination. There is no interpretability or causal evidence to support their significance.

• Lack of statistical/token-level validation: There is no human evaluation or statistical analysis demonstrating that the adaptively selected tokens indeed localize hallucinated content better than baseline tokens.

---

> ### Author Rebuttal · Authors · 2025-07-31
>
> Thank you for your thoughtful suggestions. We sincerely appreciate your positive remarks on our contributions. Please see our response to your comments one by one below.
>
> > **Question #1 \& Weakness #2**: How accurate are GPT-4.1 labels in practice? Could you report inter-agreement between the two judgment passes? Have you considered incorporating human annotations or knowledge-based QA datasets (e.g., where gold answer is guaranteed factual)?
>
> To evaluate the correctness of free-form generations in QA tasks, GPT-4 labels have been widely adopted in recent hallucination detection studies, such as Semantic Entropy [Ref1] and TSV [Ref2], establishing a popular evaluation protocol with GPT-4 used as annotator in this field. In our work, we utilize updated GPT-4.1 as the labelling model.
> We also observed that, when provided with gold answers, GPT-4.1 would give highly reliable annotation results for LLM-generated answers.
> The inter-agreements between the two judgment passes are 96.1\%, 82.3\%, 91.2\% and 92.1\%, respectively for the datasets Trivia QA, SQuAD, NQ Open and BioASQ based on LLaMA-3.1-8B model, indicating a high level of agreement across datasets. In our experiments, we exclude samples where the two GPT-4.1 judgments are inconsistent, ensuring our evaluation is based on confident, aligned labels.
>
> We also agree with the value of incorporating human annotations or leveraging knowledge-grounded QA datasets with gold factual references. To validate HaMI’s performance on such datasets, we conducted an additional experiment on TruthfulQA dataset [Ref3], which provides ground-truth true and false answers.
> HaMI achieves an AUROC score of 0.927 on this dataset, which is higher than SAPLMA (0.903) and HaloScope (0.685), as we tested. While human annotation provides high labelling fidelity, its high cost limits feasibility at scale. Due to time limitations, we haven't managed to obtain the results of all competing methods; we will include the full results on this dataset to provide empirical justification for scenarios with human-annotated labels.
>
> > **Weakness #1**: Limited baseline comparison: Important recent hallucination detection methods such as LLM-Check and CED (Comparing Embedding Differences) are not included. These would provide stronger empirical context and help position the contribution more rigorously.
>
> We appreciate your suggestion and include the suggested two additional baselines LLM-Check and CED for a more comprehensive comparison. The comparison results across all four datasets (i.e., TriviaQA, SQuAD, NQ, and BioASQ) with LLaMA-3.1-8B are summarized in **Table A1** below. Observed from the original paper on LLM-Check, we choose hidden state and attention-based variations, which claim stronger performance. As presented in Table A1, our method consistently outperforms these added baselines by a significant margin.
>
> In addition, as suggested by Reviewer **NwzN**, we have also added another recent method G-NLL [Ref4] as our baseline, and our method also obtains substantially better performance than G-NLL. Please refer to our response to Reviewer **NwzN**'s **Question 4 \& Weakness #3** for more details.
>
> | Method                   | Trivia QA | SQuAD | NQ    | BioASQ |
> |--------------------------|-----------|--------|-------|--------|
> | LLM-Check-Hidden-Score   | 0.666     | 0.614  | 0.610 | 0.673  |
> | LLM-Check-Attention-Score| 0.651     | 0.644  | 0.626 | 0.675  |
> | CED                      | 0.745     | 0.776  | 0.710 | 0.695  |
> | HaMI*      | 0.854     | 0.783  | 0.788 | 0.823  |
> | HaMI       | 0.897     | 0.826  | 0.820 | 0.836  |
> ```
> Table A1. Results for LLM-Check, CED and HaMI
> ```
>
>
> > **Weakness #3 \& Weakness #4**:
> > - Implicit token selection: The MIL approach does not guarantee that selected high-scoring tokens are truly the ones responsible for hallucination. There is no interpretability or causal evidence to support their significance.
> > - Lack of statistical/token-level validation: There is no human evaluation or statistical analysis demonstrating that the adaptively selected tokens indeed localize hallucinated content better than baseline tokens.
>
> Thank you for raising this important point regarding the validation of our token selection module.
> HaMI relies on coarse supervision signals, i.e., sequence-level labels rather than token-level labels. As a result, the MIL-based approach cannot guarantee that the tokens with the highest scores always perfectly align with the exact hallucinated words. However, we would like to clarify that although the identified token may not always correspond precisely to the hallucinated token, it often appears close, for example, immediately before or after if such  inaccurate cases happen. Moreover, hallucinated tokens themselves tend to receive scores very close to 1.0, indicating strong relevance even in the absence of fine-grained labels. This property allows HaMI to effectively narrow down the region where hallucinated content is likely to occur, which is practically useful for interpretability.
>
> To further support the validity of the token selection process, we conducted an additional study, a manual evaluation using 100 randomly sampled examples from the Trivia QA dataset. In this analysis, we manually annotated hallucinated tokens and measured the token detection recall rate of our method throughout training. Specifically, HaMI can achieve a recall rate at 0.84 as training progresses. This suggests that our approach enables the model to identify salient tokens even without token-level supervision.
>
> Please kindly refer to our responses to Reviewer **NwzN**'s **Question #1**, **Weakness #4**, and Reviewer **dBF6**'s **Question #2 \& Weakness #3** for detailed discussion and empirical evidence on the quality of the selected high-scoring tokens.
>
> **References:**
> - [Ref1] Farquhar, Sebastian, Jannik Kossen, Lorenz Kuhn, and Yarin Gal. "Detecting hallucinations in large language models using semantic entropy." Nature 630, no. 8017 (2024): 625-630. 2024
> - [Ref2] Park, Seongheon, Xuefeng Du, Min-Hsuan Yeh, Haobo Wang, and Yixuan Li. "Steer LLM Latents for Hallucination Detection." Forty-Second International Conference on Machine Learning. 2025.
> - [Ref3] Lin, Stephanie, Jacob Hilton, and Owain Evans. "Truthfulqa: Measuring how models mimic human falsehoods." 60th Annual Meeting of the Association for Computational Linguistics. 2022.
> - [Ref4] Lukas Aichberger, Kajetan Schweighofer, and Sepp Hochreiter. "Rethinking uncertainty estimation in natural language generation." arXiv preprint arXiv:2412.15176, 2024.

---

> > ### Comment · Reviewer_tguY · 2025-08-02
> > **detection metrics**
> >
> > Thank you for your detailed response, which has addressed most of my concerns. However, I would like to raise a key remaining issue regarding the need for more definitive and deterministic evaluation metrics for hallucination detection.
> >
> > While we acknowledge the increasing popularity of GPT-based or human-annotated labels for evaluation—and appreciate your reporting of high inter-annotator agreement using GPT-4.1—we believe that such approaches still carry inherent uncertainty and potential subjectivity. Even in your additional experiment referencing Park et al. (2025), “Steer LLM Latents for Hallucination Detection,” the evaluation relied on a BLEURT-based thresholding mechanism to determine ground-truth labels, rather than fully deterministic factual grounding.
> >
> > In light of this, I suggest incorporating an additional evaluation experiment that follows a more deterministic or knowledge-grounded protocol, similar to BLEURT + threshold, or even better and reporting metrics such as AUROC or accuracy under that setting? Such an experiment would offer a more objective foundation to support the reliability of the proposed hallucination detection method.

---

> ### Author Response · Authors · 2025-08-04
> **clarifications and additional experiments in response to concerns about detection metrics**
>
> Thank you very much again for your thoughtful feedback. Below, we provide additional results and clarifications in response to your suggestion.
>
> Following your suggestion, we conducted an additional experiment, using the BLEURT-threshold method to generate labels and comparing HaMI with most baselines (due to time constraints). Results are reported in Table A2 below, which shows that our methods HaMI and HaMI* can also consistently outperform the baselines using the BLEURT-based labelling datasets. Comparing to the GPT labels-based AUROC results in Table 1 in our paper, the internal state-based methods, including CCS, SAPLMA, HaloScope, and our method HaMI*, generally obtain largely improved AUROC performance based on the BLEURT labels, whereas uncertainty-based methods Perplexity and SE perform not that stably, i.e., performing better on some of the datasets while being less effective on the other datasets. Although HaMI (the last row in Table A2) includes uncertainty semantic-based feature enhancement, our proposed MIL hallucination score learning helps well balance the internal states and the uncertainty semantic, achieving consistently better AUROC performance than HaMI* in Table A2. These results provide empirical evidence for the robustness of HaMI from another perspective (a data labelling perspective).
>
> | Method      | Trivia QA | SQuAD  | NQ     | BioASQ |
> |-------------|-----------|--------|--------|--------|
> | Perplexity  | 0.663     | 0.638  | 0.586  | 0.662  |
> | SE          | 0.709     | 0.813  | 0.704  | 0.714  |
> | CCS         | 0.682     | 0.657  | 0.738  | 0.683  |
> | SAPLMA      | 0.897     | 0.898  | 0.867  | 0.873  |
> | HaloScope   | 0.792     | 0.845  | 0.852  | 0.830  |
> | HaMI*       | 0.900     | 0.928  | 0.909  | 0.918  |
> | **HaMI**    | **0.909** | **0.937** | **0.920** | **0.922** |
>
> ```
> Table A2: AUROC results with labels generated by the BLEURT method. The results are based on LLaMA-3.1-8B.
> ```
>
> Nevertheless, we would like to note that, as we manually checked the annotation results, the BLEURT-threshold method is found better suited for phrase-level LLM responses, which often yield unexpected incorrect labels for sentence-level responses.
> For example, for the question:
> *“In April, which sportsman married his childhood sweetheart Kim Sears?”*
> The given ground truth answer is *"Andy Murray"* in the dataset. BLEURT labelled *“Andy Murray.”* as correct, but incorrectly labelled the sentence *“In April, tennis player Andy Murray married his childhood sweetheart Kim Sears.”* as incorrect. This type of failure is likely due to its sensitivity to answer length and format.
>
> Besides, we noticed that the referenced Park et al. (2025) also report GPT-based evaluation results in Appendix C.2.
> Based on your suggestion and the above observations, we will include the results of all baselines under the BLEURT setting in our final version to complement the GPT-based evaluation results, and discuss the potential strengths and weaknesses under the GPT/BLEURT-based evaluation.
>
> Thank you again for the insightful suggestion. We hope these replies help address your remaining concern. We are more than happy to engage with you to discuss any other questions you may have.

---

> > ### Comment · Reviewer_tguY · 2025-08-06
> >
> > I'm satisfied with the responses - I don't think there are any outstanding questions.  I will raise my rating.

---

> > > ### Author Response · Authors · 2025-08-06
> > >
> > > Thank you very much for your feedback. We truly appreciate your support and are glad that our responses addressed your concerns.

---

### Official Review · Reviewer_dBF6 · 2025-07-03

**Clarity:** 4
**Significance:** 3
**Originality:** 3
**Rating:** 5
**Confidence:** 3

**Summary:**

This paper proposes HaMI (Hallucination detection as Multiple Instance learning), a novel approach for detecting hallucinations in large language models through adaptive token selection. The key innovation is reformulating hallucination detection as a multiple instance learning problem where each response sequence is treated as a bag of token instances, enabling joint optimization of token selection and hallucination detection. The method adaptively identifies the most indicative tokens for hallucination rather than relying on predetermined token positions (first, last, etc.). Additionally, the authors introduce a representation enhancement module that integrates uncertainty information at multiple levels (token, sentence, and semantic consistency) to improve detection performance. Experimental results on four QA benchmarks (TriviaQA, SQuAD, Natural Questions, BioASQ) with three LLMs (LLaMA-3.1-8B, Mistral-Nemo-12B, LLaMA-3.13-70B) demonstrate significant improvements over state-of-the-art methods.

**Questions:**

- Can the authors provide more empirical analysis of when the smoothness assumption holds? Are there cases where abrupt changes in hallucination scores between adjacent tokens are actually beneficial (or would not degrade the performance much)?
- What are the main failure modes of HaMI? When does adaptive token selection fail to identify the most relevant tokens?
- Do different types of hallucinations (factual errors, nonsensical responses, contradictions) show different patterns in token selection? How does HaMI perform on fine-grained hallucination taxonomies?

**Ethical Concerns:**

["NO or VERY MINOR ethics concerns only"]

**Limitations:**

Yes

**Quality:**

3

**Strengths And Weaknesses:**

Strengths:
- Novel Problem Formulation: The MIL reformulation is intuitive and well-motivated. The assumption that only a few tokens contain hallucination information while most tokens in incorrect responses are neutral aligns well with the sparse nature of hallucinations.
- End-to-End Joint Optimization: Unlike existing methods that separate token selection and detection into two stages, HaMI enables joint optimization, which is a methodological advancement.
- Comprehensive Experimental Evaluation: The experiments cover multiple datasets, models, and baselines.
- Strong Empirical Results: HaMI achieves substantial improvements over baselines, with gains of 8.1% to 11.9% AUROC over MARS-SE across three LLMs.
- Thorough Ablation Studies: The paper provides detailed analysis of different components, uncertainty types, layer selection, and hyperparameter sensitivity.

Weaknesses:
- The best-performing variant (HaMI with semantic consistency) requires multiple generations and entailment evaluation, which significantly increases computational cost (7.6-46.1 seconds per question as noted in appendix).
- The sequential smoothness loss (Eq. 3) assumes neighbor tokens should have similar hallucination scores, but this assumption may not always hold in practice (e.g., when hallucinations occur at specific entity mentions).
- The paper misses discussion of when and why the method fails, which would be valuable for understanding its limitations.

---

> ### Author Rebuttal · Authors · 2025-07-31
>
> We are grateful for your insightful comments and encouraging evaluation of our work. Below, we provide detailed responses to each of your comments.
>
> > **Question #1 \& Weakness #2**: Can the authors provide more empirical analysis of when the smoothness assumption holds? Are there cases where abrupt changes in hallucination scores between adjacent tokens are actually beneficial (or would not degrade the performance much)?
>
> Thank you very much for the question. The smoothness loss is designed under the assumption of continuity in hallucination scores between adjacent tokens. As shown by the added ablation study results in **Tables A1 and A2** below, this assumption holds well in practice since adding the smoothness loss consistently leads to improved or comparable performance across multiple benchmarks;  notably, the improvement is more pronounced in the cross-dataset setting, indicating its positive role in serving as a regularization term for better generalization across the domains. Please refer to our response to Reviewer **NwzN**'s **Question #2** for more detailed discussion of these ablation study results.
>
> | $\mathcal{L}_{MIL}$ | $\mathcal{L}_{Smooth}$ | Trivia QA | SQuAD | NQ    | BioASQ |
> |-----------|------|-----------|-------|-------|--------|
> | ✓         |      | 0.892±0.005 |0.825±0.002 | 0.812±0.003 | 0.840±0.015|
> | ✓         | ✓    | 0.898±0.007 |0.835±0.006 | 0.815±0.004 | 0.845±0.012 |
> ```
> Table A1. Ablation study on the two proposed losses (AUROC).
> ```
>
> | $\mathcal{L}_{MIL}$ | $\mathcal{L}_{Smooth}$ | Trivia QA | SQuAD | NQ    | BioASQ |
> |-----------|----|-----------|-------|-------|--------|
> | ✓         |    | 0.852±0.012  | 0.812±0.005 | 0.797±0.020 | 0.790±0.007  |
> | ✓         | ✓  | 0.865±0.008  | 0.820±0.002 | 0.808±0.012 | 0.795±0.002  |
> ```
> Table A2. Ablation study on the two proposed losses under the cross-dataset setting.
> ```
>
> The smooth hallucination scores can enhance the optimization due to the following two main reasons. i) For non-hallucination sequences, all tokens are non-hallucination ones, and enforcing the smoothness over them results in consistently small hallucination scores for them (see Figure 4 for an example); for hallucination sequences, the smoothness loss does not stop the hallucination tokens from gaining large hallucination scores, but it can increase the scores of the neighborhood tokens of the hallucination tokens, i.e., to have a gradual rise and down at around the hallucination tokens while having consistently small scores for the non-hallucination tokens. This is typically observed in the experimental datasets. Additionally, since we pick the top-k tokens based on the hallucination scores of the tokens, the hallucinated tokens would be chosen as long as the margin between the scores of hallucinated and non-hallucinated tokens is sufficiently large; the token selection results would be similar to the cases that we have very large margins under the abrupt changes in hallucination scores. ii) The smoothness loss also effectively regularizes the model to avoid excessive attention to suspicious tokens, helping avoid overfitting the training data, and thereby enabling better generalization.
>
> > **Question #2 \& Weakness #3**: What are the main failure modes of HaMI? When does adaptive token selection fail to identify the most relevant tokens?
>
> One primary limitation of HaMI lies in its coarse supervision signals, i.e., the token selection in HaMI relies on sequence-level class labels. Since we do not have fine-grained, token-level hallucination labels, HaMI may sometimes fail to attribute the exact hallucinated tokens with the highest scores. Nonetheless, while the hallucination scores to the true hallucinated tokens may not be the absolute highest value, these tokens often still receive scores close to 1.0, indicating strong relevance. Moreover, although the highest score is not with the exact hallucinated token, it typically appears before or after the exact token. As such, HaMI effectively narrows the region where hallucinated content is likely to occur. This localization ability is valuable in practical settings, as it supports interpretability and downstream correction mechanisms, even in the absence of perfect token-level attribution.
>
> We also notice that the effectiveness of the token selection module may be limited if the generated sequence is short, such as phrase-level generations. In these instances, the entire sequence is typically related to the exact hallucination, and the pre-defined final token may encapsulate the concentrated semantic better.
>
> Thank you for your thoughtful question and we will include the discussion in our paper.
>
>
> > **Question #3**: Do different types of hallucinations (factual errors, nonsensical responses, contradictions) show different patterns in token selection? How does HaMI perform on fine-grained hallucination taxonomies?
>
> Thank you very much for the question. In the current work, we did not explicitly evaluate HaMI on fine-grained hallucination taxonomies such as factual errors, nonsensical responses, or contradictions. Our primary evaluation focused on general hallucination detection. However, we conducted an empirical analysis using the TruthfulQA dataset [Ref1], which includes class labels for different hallucination types. Due to time limitations, we did not separate the training set by hallucination type;  HaMI is optimized on the training data with all hallucination types. At inference time, we included all response types and observed that HaMI's performance was relatively consistent across different hallucination categories. Hope this helps address your concern.
>
> > **Weakness #1**: The best-performing variant (HaMI with semantic consistency) significantly increases computational cost.
>
> We include semantic entropy as uncertainty enhancement feature into HaMI because it provides a strong signal for hallucination detection. However, we agree that integrating the sementic entropy will introduce additional overhead due to the generation of multiple outputs and entailment evaluation.
> To eliminate the influence on computational efficiency, we also evaluated HaMI*, which relies on only a single generation.
>
> As shown in Table 1 in our paper, HaMI* achieves consistently superior performance on all four datasets, compared to all other single-generation-based baselines, including strong methods like SAPLMA and MARS. Even when compared with multiple-generation-based approaches such as SE, HaMI* outperforms or matches the best results across datasets, demonstrating that our method is not only effective but also adaptable to resource-constrained settings.
>
> **References:**
> - [Ref1] Lin, Stephanie, Jacob Hilton, and Owain Evans. "Truthfulqa: Measuring how models mimic human falsehoods." 60th Annual Meeting of the Association for Computational Linguistics. 2022.

---

### Official Review · Reviewer_NwzN · 2025-07-06

**Clarity:** 2
**Significance:** 3
**Originality:** 3
**Rating:** 4
**Confidence:** 4

**Summary:**

The paper introduces Hallucination detection as Multiple Instance learning, short HaMI, which trains a hallucination detector that distinguishes between generations with hallucinations (positive bags of tokens) and generations without hallucinations (negative bags of tokens) by maximizing the distance between them in token-level representation space.

**Questions:**

- The authors "choose the token instances with the top k highest hallucination scores as the salient tokens in each sequence" (line 156). These hallucination scores are obtained from the hallucination detector. However, at the beginning of training, these are random scores, so maximizing the discriminative margin between these randomly selected tokens might not be optimal. How does this influence training?
- How does the sequential smoothness loss impact the training and performance of the method? (Re. Weakness 1)
- Would HaMI still achieve the best performance in Table 1 if a single hallucination detector were trained per model, rather than per dataset? (Re. Weakness 2)
- How does the method compare to single-sequence uncertainty method both in performance and computational effectiveness? (Re. Weakness 3)

**Ethical Concerns:**

["NO or VERY MINOR ethics concerns only"]

**Final Justification:**

My questions and concerns have largely been resolved in the authors' rebuttal.

The authors confirmed that:
- HaMI's sequential smoothness loss improves performance, even if only marginally.
- HaMI remains effective when the detector is trained on a single dataset and evaluated on the other three.
- HaMI improves upon single-sequence uncertainty methods.

Overall, the rebuttal strengthened and clarified the contributions. I have therefore increased my score.

**Limitations:**

yes

**Paper Formatting Concerns:**

No major formatting issues noticed.

**Quality:**

3

**Strengths And Weaknesses:**

**Strengths:**
1. The paper proposes a an interesting approach to hallucination detection by jointly optimizing token selection and classification via a Multiple Instance Learning (MIL) formulation.
2. HaMI demonstrates strong and consistent empirical performance, significantly outperforming state-of-the-art baselines across both uncertainty-based and internal-representation-based detection methods.
3. The evaluation is thorough and well-structured, spanning three language models and four QA datasets.
4. The ablation studies are well-designed and clearly show the contribution of each component to the overall performance.

**Weaknesses:**
1. The authors introduce a sequential smoothness loss based on the assumption that hallucination scores for neighboring tokens should be similar. However, this assumption has not been theoretically motivated or empirically validated. It would be helpful to further investigate this assumption.
2. For the main results, the hallucination detector is both trained and evaluated on the same dataset, whereas the uncertainty estimation baselines are dataset-agnostic. A fairer comparison would be to train a single detector per model and evaluate it across multiple datasets. While Section 5.2 provides some insight into HaMI’s generalization ability, it remains unclear how its performance compares to that of the uncertainty estimation baselines. Additionally, it is not clear whether the method generalizes beyond the question answering domain.
3. The baselines do not include sentence-level uncertainty measures that require generating only a single output sequence, such as G-NLL, which is computed using the most-likely output sequence only [1]. These methods are lightweight and widely applicable, thus should be considered standard in an evaluation.
4. The authors claim that their method can identify the most hallucinated tokens within the positive bag. However, the ablation in Section 5.3 shows that semantically irrelevant words like “is” and “and” receive the highest hallucination scores..

---
[1] Lukas Aichberger, Kajetan Schweighofer, and Sepp Hochreiter. Rethinking uncertainty estimation in natural language generation. *arXiv preprint arXiv:2412.15176*, 2024.

---

> ### Author Rebuttal · Authors · 2025-07-31
>
> Thank you very much for your valuable and detailed feedback. We are grateful for your positive assessment of our technical contributions and empirical results. Please see our response to your comments one by one below.
>
> > **Question #1** How does randomly choosing tokens early in training affect the process?
>
> Thank you very much for the question. The optimization in HaMI is analogous to an alternating-optimization approach, in which one of the two sub-problems, token selection and hallucination detection, is alternatively solved, with the parameters in the other one fixed. Please note that, to simplify the optimization, there are no learnable parameters in token selection; we select the tokens with the largest hallucination scores.
>
> It is true that at the beginning of training, the token hallucination scores predicted by the detector are not properly optimized, and therefore, the selected tokens based on these scores may be not that relevant to hallucination detection.
> However, as more optimization epochs are done, the detector becomes more and more optimal, resulting in better token-level hallucination scores and thus better token selection, and vice versa.
> Specifically, as training progresses on the datasets used in our experiments, the hallucination scores for tokens in positive bags increase gradually from below 0.5 (observed from the first 10 training steps) to values approaching 1.0, indicating stronger confidence in detecting hallucinated tokens. This phenomenon is observed on all our datasets and is consistently stable with different random parameter initialization seeds.
>
> To further examine the effectiveness of token selection, we randomly selected 100 samples from the Trivia QA dataset and manually annotated the hallucinated tokens. We then tracked the recall rate in the token selection module at different stages of training. We observed that the recall rate of detecting the hallucinated tokens initially was relatively quite high (0.72 after just two training steps), then dropped to around 0.66 for several steps, and steadily improved and became stable thereafter, flattening at a recall rate of about 0.84. This recall rate is remarkable since the full training process is done without using token-level annotations.
>
> These observations suggest that although the top-k token selection strategy may be random in the beginning, the MIL-based joint optimization framework enables HaMI to progressively refine its token selection and effectively learn to identify accurate hallucinated tokens. We will add these results and clarification to the final version to address this concern.
>
> > **Question #2 \& Weekness #1**: How does the sequential smoothness loss impact the training and performance of the method?
>
> Thank you very much for raising this question. We added ablation study on four benchmarks with LLaMA-3.1-8B model for the proposed loss functions, including both MIL loss and smoothness loss. Main results on each individual dataset and cross-dataset results are respectively presented in **Tables A1** and **A2** below. The results are averaged over three independent runs. As evidenced in the tables, incorporating the smoothness loss consistently yields improved performance across all the cases. On average, the smoothness loss consistently enhances HaMI, increasing the performance by 0.7\% and 1.1\% on the individual-dataset and cross-dataset scenarios, respectively.
> This demonstrates that the contribution of the smoothness loss is more pronounced in the more challenging cross-dataset setting. These consistent improvements highlight that the smoothness loss plays a rather positive role in helping achieve a more robust and generalizable HaMI, which is particularly valuable in the cross-domain scenarios. We will add these results and this discussion in the final version to clarify the role of this loss.
>
>
> | $\mathcal{L}_{MIL}$ | $\mathcal{L}_{Smooth}$ | Trivia QA | SQuAD | NQ    | BioASQ |
> |-----------|------|-----------|-------|-------|--------|
> | ✓         |      | 0.892±0.005 |0.825±0.002 | 0.812±0.003 | 0.840±0.015|
> | ✓         | ✓    | 0.898±0.007 |0.835±0.006 | 0.815±0.004 | 0.845±0.012 |
> ```
> Table A1. Ablation study on the two loss functions (AUROC).
> ```
>
> | $\mathcal{L}_{MIL}$ | $\mathcal{L}_{Smooth}$ | Trivia QA | SQuAD | NQ    | BioASQ |
> |-----------|----|-----------|-------|-------|--------|
> | ✓         |    | 0.852±0.012  | 0.812±0.005 | 0.797±0.020 | 0.790±0.007  |
> | ✓         | ✓  | 0.865±0.008  | 0.820±0.002 | 0.808±0.012 | 0.795±0.002  |
>
> ```
> Table A2. Ablation study on the two loss functions (AUROC) under the cross-dataset setting.
> ```
>
>
>
> > **Question #3 \& Weakness #2**: Would HaMI still achieve the best performance in Table 1 if a single hallucination detector were trained per model, rather than per dataset?... it is not clear whether the method generalizes beyond the question answering domain.
>
> Thank you very much for the question. And yes, as shown in Figure 3(a) in our paper,  HaMI maintains superior performance when it is trained per model rather than per dataset, i.e., the detector is trained once on one single dataset and then evaluated on all the other three datasets. This setting is referred to as cross-dataset generalization in the paper. Here we present the experimental results with different settings on four datasets based with the LLaMA-3.1-8B model in **Table A3**.
> For comparison, we also present the best results on each dataset from all non-learnable/training-free baselines: p(True), Perplexity, SE, MARS, MARS-SE (extracted from Table 1 of our paper), all of which consistently underperform HaMI across the datasets.
>
>
> |                                | Trivia QA     | SQuAD        | NQ           | BioASQ       |
> |--------------------------------|---------------|--------------|--------------|--------------|
> | HaMI - cross-dataset-performance | 0.865±0.008  | 0.820±0.002 | 0.808±0.012 | 0.795±0.002  |
> | Best AUROC for non-learnable methods | 0.828   | 0.787        | 0.777        | 0.757        |
>
> ```
> Table A3. Cross-dataset generalization capability compared with non-learnable/training-free baselines.
> ```
>
>
> For your concern on HaMI's capability beyond QA domain in **Weakness #2**, we would like to clarify that HaMI can be directly employed in summarization tasks, where the target text and task instruction serve as the context prompt and question, respectively, and the summary is treated as the answer. For other tasks, such as consistency checking in storytelling, we can also reformulate them into a QA format for evaluation.
>
>
> > **Question #4 \& Weakness #3**: How does the method compare to single-sequence uncertainty method both in performance and computational effectiveness?
>
> Following your suggestion, we include the single-sequence uncertainty method G-NLL [Ref1] as a baseline method. Since official implementation code is not available, we follow the methodology described in the original paper to reproduce their method. All experiments were conducted using the LLaMA-3.1-8B model across four benchmark datasets.
>
> As shown in **Table A4** below, HaMI significantly outperforms G-NLL across all four datasets, with the improvement ranging from 20\% to 30\%. Notably, even HaMI*, a variant of HaMI that uses only a single generation, can substantially outperform G-NLL, further validating the superior performance of our formulation over G-NLL.
> The other baseline *Perplexity*, with its performance reported in Table 1 in our paper, also relies on sentence-level aggregation of token-level negative log-likelihoods. Its performance shows similar trends as G-NLL.
>
> In terms of computational efficiency, HaMI requires additional overhead due to the need to process high-dimensional representations and the training of the detector. In contrast, the single-sequence methods like G-NLL only requires predictive logits and does not involve additional training, making it computationally more efficient than HaMI.
> However, HaMI remains practical since the inference process operates at the token level without requiring large computational costs. We believe this trade-off between computational cost and substantial accuracy gains justifies the approach.
>
> |           Method | Trivia QA | SQuAD | NQ    | BioASQ |
> |------------------|-----------|--------|-------|--------|
> | G-NLL            | 0.732     | 0.683  | 0.612 | 0.692  |
> | HaMI*            | 0.854     | 0.783  | 0.788 | 0.823  |
> | HaMI             | 0.897     | 0.826  | 0.820 | 0.836  |
>
> ```
> Table A4. Results for G-NLL and HaMI
> ```
>
> > **Weakness #4**: Concerns regarding the ablation study results on token identification.
>
> Thank you very much for the question. As mentioned in our paper, HaMI operates without access to token-level supervision. Consequently, while the model aims to identify tokens associated with hallucinated content, it relies on weak supervision signals derived from sequence/bag-level labels. This inevitably introduces some "noise". It means that the highlighted tokens may not always align precisely with the hallucinated words, but may appear before or after them, which can be meaningful for locating where hallucination appears. We have discussed this phenomenon in Section 5.3 (Lines 306-313). Nevertheless, as we discussed in our response to **Question #1**, given that the optimization of HaMI does not involve any token-level label information, its recall rate of detecting the true hallucinated tokens (~0.85) is rather high, offering strong supervisory signals for building an accurate hallucination detector.
>
> **References:**
> - [Ref1] Lukas Aichberger, Kajetan Schweighofer, and Sepp Hochreiter. "Rethinking uncertainty estimation in natural language generation." arXiv preprint arXiv:2412.15176, 2024.

---

> > ### Comment · Reviewer_NwzN · 2025-08-05
> >
> > Thank you for the clarifications and for conducting the ablation studies. I encourage the authors to incorporate these additional insights, as they will strengthen the paper and improve its overall quality. I do not have any further questions and am inclined to recommend acceptance.

---

> > > ### Author Response · Authors · 2025-08-05
> > >
> > > We sincerely thank you for the constructive comments and encouraging recommendation. We appreciate your support and will incorporate the additional insights and experimental results into the final version to further strengthen the paper.

---

### Note · Authors · 2025-08-14

Dear Reviewers, AC and Senior AC,

We sincerely thank you for your dedication and feedback throughout the review process. Below, we provide a summary of the key issues raised during the discussion phase and how we addressed them.

>**Top-k token selection.**

All the reviewers raised concerns about the effectiveness of MIL-based top-k token selection strategy.
To address this, we manually labeled a set of hallucinated tokens and tracked the recall rate of identified tokens during training. The results demonstrate that HaMI can accurately identify salient tokens despite being trained only with sequence-level supervision.

>**Effectiveness of the smoothness loss.**

Reviewers NwzN and dBF6 questioned the contribution of the smoothness loss. Additional ablations show that it consistently improves detection performance.

>**Generalization capabilities.**

Reviewers NwzN and Z1J4 were concerned the generalization capability of HaMI. Our cross-dataset evaluations demonstrate that HaMI consistently outperforms baselines even in non-learnable settings.

>**Missing baselines.**

Reviewers NwzN and tguY suggested additional baselines. We conducted experiments with these baselines and showed that HaMI consistently outperforms them.

>**Label quality.**

Concerns were raised about the reliability of GPT-4.1 annotations. In response, we evaluated HaMI on datasets with verified true/false labels and BLEURT-based annotations separately. In both settings, HaMI outperformed baselines.

>**Computation efficiency.**

Concerns were raised about the computational cost of the SE-enhanced HaMI model. We clarified that HaMI*, a lightweight variant requiring only a single generation, maintains superior performance while being more efficient.

>**Failure cases.**

Reviewers dBF6 asked about the main failure case of HaMI. We acknowledged that using sequence-level supervision may lead to some inaccurate attribution of hallucinated tokens. However, HaMI enables effective localization despite the lack of token-level labels.

In conclusion, during the discussion phase, we carefully addressed the reviewers' concerns with additional experiments, ablations, and clarifications. We are grateful for the reviewers’ constructive engagement and would like to express our sincere thanks once again for the reviewers’ positive support for the acceptance of our paper.

Best regards,
Authors

---

### Decision · Program_Chairs · 2025-09-17

**Decision:**

Accept (poster)

**Comment:**

This paper introduces HaMI, a novel approach for LLM hallucination detection that reformulates the problem as Multiple Instance Learning (MIL) with adaptive token selection, receiving consistent support from all reviewers. The core innovation lies in treating response sequences as "bags" of token instances and jointly optimizing token selection and classification, moving beyond predetermined token positions used in prior work. Reviewers praised the intuitive MIL formulation and strong empirical results. Authors provided comprehensive rebuttals including manual token annotation studies, extensive baseline comparisons and clarification that the lightweight variant maintains superior performance while being computationally efficient. All reviewers ultimately supported acceptance, recognizing this as a solid incremental contribution that opens new research directions in hallucination detection through the MIL framework.